# DR-DSGD: A Distributionally Robust Decentralized Learning Algorithm over Graphs

**Chaouki Ben Issaid**                                        *chaouki.benissaid@oulu.fi*
*Centre for Wireless Communications*
*University of Oulu, Finland*

**Anis Elgabli**                                              *anis.elgabli@oulu.fi*
*Centre for Wireless Communications*
*University of Oulu, Finland*

**Mehdi Bennis**                                             *mehdi.bennis@oulu.fi*
*Centre for Wireless Communications*
*University of Oulu, Finland*

**Reviewed on OpenReview:** *https://openreview.net/forum?id=VcXNAr5Rur*

## Abstract

In this paper, we propose to solve a regularized distributionally robust learning problem in the decentralized setting, taking into account the data distribution shift. By adding a Kullback-Liebler regularization function to the robust min-max optimization problem, the learning problem can be reduced to a modified robust minimization problem and solved efficiently. Leveraging the newly formulated optimization problem, we propose a robust version of Decentralized Stochastic Gradient Descent (DSGD), coined Distributionally Robust Decentralized Stochastic Gradient Descent (DR-DSGD). Under some mild assumptions and provided that the regularization parameter is larger than one, we theoretically prove that DR-DSGD achieves a convergence rate of $\mathcal{O}\left(1/\sqrt{KT} + K/T\right)$, where $K$ is the number of devices and $T$ is the number of iterations. Simulation results show that our proposed algorithm can improve the worst distribution test accuracy by up to 10%. Moreover, DR-DSGD is more communication-efficient than DSGD since it requires fewer communication rounds (up to 20 times less) to achieve the same worst distribution test accuracy target. Furthermore, the conducted experiments reveal that DR-DSGD results in a fairer performance across devices in terms of test accuracy.

## 1 Introduction

Federated learning (FL) is a learning framework that allows the training of a model across multiple devices under the orchestration of a parameter server (PS). Unlike the traditional way of training ML models, where the individual data of the devices are shared with the PS, FL hides the raw data, so it can significantly reduce the communication cost. When combined with a privacy-preservation mechanism, it further ensures privacy. Training using FedAvg presents several challenges that need to be tackled. It often fails to address the data heterogeneity issue. Though many enhanced variants of federated averaging (FedAvg) (Li et al., 2020b; Liang et al., 2019; Karimireddy et al., 2020b;a; Wang et al., 2020; Mitra et al., 2021; Horváth et al., 2022) were proposed to solve this issue in the mean risk minimization setting, local data distributions might differ greatly from the average distribution. Therefore, a considerable drop in the global model performance on local data is seen, suggesting that the mean risk minimization may not be the right objective. Another major issue in FL is fairness. In many cases, the resultant learning models are biased or unfair in the sense that they discriminate against certain device groups (Hardt et al., 2016). Finally, FL relies on the existence

of a PS to collect and distribute model parameters, which is not always feasible or even accessible to devices that are located far away.

Even though several FL algorithms (Kairouz et al., 2021; Yang et al., 2019; Li et al., 2020a) have been proposed for the distributed learning problem, FedAvg-style methods (McMahan et al., 2017; Li et al., 2020b; Wang et al., 2020) remain the state-of-the-art algorithm. Specifically, FedAvg entails performing one or multiple local iterations at each device before communicating with the PS, which in turn performs periodic averaging. However, because FedAvg is based on the empirical risk minimization (ERM) to solve the distributed learning problem, i.e. FedAvg minimizes the empirical distribution of the local losses, its performance deteriorates when the local data are distributed non-identically across devices. While the ERM formulation assumes that all local data come from the same distribution, local data distributions might significantly diverge from the average distribution. As a result, even though the global model has a good average test accuracy, its performance locally drops when the local data are heterogeneous. In fact, increasing the diversity of local data distributions has been shown to reduce the generalization ability of the global model derived by solving the distributed learning problem using FedAvg (Li et al., 2020e;c; Zhao et al., 2018).

While there are many definitions of robustness, the focus of our work is on distributionally robustness, that is, being robust to the distribution shift of the local data distributions. We consider a distributionally robust learning perspective by seeking the best solution for the worst-case distribution. Another key focus of this work is to investigate the fairness of the performance across the different devices participating in the learning. Fairness aims to reduce the difference in performance on the local datasets to ensure that the model performance is uniform across the devices participating in the learning process. In the FL context, achieving fair performance among devices is a critical challenge. In fact, existing techniques in FL, such as FedAvg (McMahan et al., 2017) lead to non-uniform performance across the network, especially for large networks, since they favour or hurt the model performance on certain devices. While the average performance is high, these techniques do not ensure a uniform performance across devices.

PS-based learning (star topology) incurs a significant bottleneck in terms of communication latency, scalability, bandwidth, and fault tolerance. Decentralized topologies circumvent these limitations and hence have significantly greater scalability to larger datasets and systems. In fact, while the communication cost increases with the number of devices in the PS-based topology, it is generally constant (in a ring or torus topology), or a slowly increasing function in the number of devices since decentralizing learning only requires on-device computation and local communication with neighboring devices without the need of a PS. Several works investigated the decentralizing learning problem (Yuan et al., 2016; Zeng & Yin, 2018; Wang et al., 2019; Wei & Ozdaglar, 2012; Shi et al., 2014; Ben Issaid et al., 2021; Duchi et al., 2011); however, while interesting none of these works has considered solving the decentralized learning problem in a distributionally robust manner.

**Summary of Contributions.** The main contributions of this paper are summarized as follows

- We propose a **distributionally robust learning algorithm**, dubbed Distributionally Robust Decentralized Stochastic Gradient Descent (DR-DSGD), that solves the learning problem in a decentralized manner while being robust to data distribution shift. To the best of our knowledge, our framework is the first to solve the regularized distributionally robust optimization problem in a decentralized topology.

- Under some mild assumptions and provided that the regularization parameter is larger than one, we prove that DR-DSGD achieves a fast convergence rate of $\mathcal{O}\left(\frac{1}{\sqrt{KT}} + \frac{K}{T}\right)$, where $K$ is the number of devices and $T$ is the number of iterations, as shown in Corollary 1. Note that, unlike existing FL frameworks that rely on the **unbiasedness** of the stochastic gradients, our analysis is more challenging and different from the traditional analyses for decentralized SGD since it involves **biased** stochastic gradients stemming from the compositional nature of the reformulated loss function.

- We demonstrate the robustness of our approach compared to vanilla decentralized SGD via numerical simulations. It is shown that DR-DSGD leads to an improvement of up to 10% in the worst distri-

bution test accuracy while achieving a reduction of up to 20 times less in terms of communication rounds.

- Furthermore, we show by simulations that DR-DSGD leads to a fairer performance across devices in terms of test accuracy. In fact, our proposed algorithm reduces the variance of test accuracies across all devices by up to 60% while maintaining the same average accuracy.

**Paper Organization.** The remainder of this paper is organized as follows. In Section 3, we briefly describe the problem formulation and show the difference between the ERM and distributionally robust optimization (DRO) formulation. Then, we present our proposed framework, *DR-DSGD*, for solving the decentralized learning problem in a distributionally robust manner in Section 4. In Section 5, we prove the convergence of DR-DSGD theoretically under some mild conditions. Section 6 validates the performance of DR-DSGD by simulations and shows the robustness of our proposed approach compared to DSGD. Finally, the paper concludes with some final remarks in Section 7. The details of the proofs of our results are deferred to the appendices.

## 2 Related Works

**Robust Federated Learning.** Recent robust FL algorithms (Mohri et al., 2019; Reisizadeh et al., 2020; Deng et al., 2020; Hamer et al., 2020) have been proposed for the learning problem in the PS-based topology. Instead of minimizing the loss with respect to the average distribution among the data distributions from local clients, the authors in (Mohri et al., 2019) proposed agnostic federated learning (AFL), which optimizes the global model for a target distribution formed by any mixture of the devices' distributions. Specifically, AFL casts the FL problem into a min-max optimization problem and finds the worst loss over all possible convex combinations of the devices' distributions. Reisizadeh et al. (2020) proposed FedRobust, a variant of local stochastic gradient descent ascent (SGDA), aiming to learn a model for the worst-case affine shift by assuming that a device's data distribution is an affine transformation of a global one. However, FedRobust requires each client to have enough data to estimate the local worst-case shift; otherwise, the global model performance on the worst distribution deteriorates. The authors in (Deng et al., 2020) proposed a distributionally robust federated averaging (DRFA) algorithm with reduced communication. Instead of using the ERM formulation, the authors adopt a DRO objective by formulating a distributed learning problem to minimize a distributionally robust empirical loss, while periodically averaging the local models as done in FedAvg (McMahan et al., 2017). Using the Bregman Divergence as the loss function, the authors in (Hamer et al., 2020) proposed FedBoost, a communication-efficient FL algorithm based on learning the optimal mixture weights on an ensemble of pre-trained models by communicating only a subset of the models to any device. The work of (Pillutla et al., 2019) tackles the problem of robustness to corrupted updates by applying robust aggregation based on the geometric median. Robustness to data and model poisoning attacks was also investigated by (Li et al., 2021b). DRO has been applied in multi-regional power systems to improve reliability by considering the uncertainty of wind power distributions and constructing a multi-objective function that maintains a trade-off between the operational cost and risk (Li & Yang, 2020; Hu et al., 2021). However, none of these works provides any convergence guarantees. Furthermore, our analysis is different from these works since it involves biased stochastic gradients.

**Fairness in Federated Learning.** Recently, there has been a growing interest in developing FL algorithms that guarantee fairness across devices (Mohri et al., 2019; Li et al., 2020d; 2021a). Inspired by works in fair resource allocation for wireless networks, the authors in (Li et al., 2020d) proposed $q$-FFL, an FL algorithm that addresses fairness issues by minimizing an average reweighted loss parameterized by $q$. The proposed algorithm assigns larger weights to devices with higher losses to achieve a uniform performance across devices. Tilted empirical risk minimization (TERM), proposed in (Li et al., 2021a), has a similar goal as $q$-FFL, i.e. to achieve fairer accuracy distributions among devices while ensuring similar average performance.

**Decentralized Learning.** Decentralized optimization finds applications in various areas including wireless sensor networks (Mihaylov et al., 2009; Avci et al., 2018; Soret et al., 2021), networked multi-agent systems (Inalhan et al., 2002; Ren et al., 2007; Johansson, 2008), as well as the area of smart grid implementations

(Kekatos & Giannakis, 2012). Several popular algorithms based on gradient descent (Yuan et al., 2016; Zeng & Yin, 2018; Wang et al., 2019), alternating direction method of multipliers (ADMM) (Wei & Ozdaglar, 2012; Shi et al., 2014; Ben Issaid et al., 2021), or dual averaging (Duchi et al., 2011) have been proposed to tackle the decentralized learning problem.

## 3 Notations & Problem Formulation

### 3.1 Notations

Throughout the whole paper, we use bold font for vectors and matrices. The notation $\nabla f$ stands for the gradient of the function $f$, and $\mathbb{E}[\cdot]$ denotes the expectation operator. The symbols $\|\cdot\|$, and $\|\cdot\|_F$ denote the $\ell_2$-norm of a vector, and the Frobenius norm of a matrix, respectively. For a positive integer number $n$, we write $[n] \triangleq \{1, 2, \ldots, n\}$. The set of vectors of size $K$ with all entries being positive is denoted by $\mathbb{R}_+^K$. The notations $\mathbf{0}$ and $\mathbf{1}$ denote a vector with all entries equal to zero, or one, respectively (its size is to be understood from the context). Furthermore, we define the matrices: $\boldsymbol{I}$ the identity matrix and $\boldsymbol{J} = \frac{1}{K}\mathbf{1}\mathbf{1}^T$. For a square matrix $\boldsymbol{A}$, $\mathrm{Tr}(\boldsymbol{A})$ is the trace of $\boldsymbol{A}$, i.e. the sum of elements on the main diagonal. Finally, for the limiting behavior of functions, $f = \mathcal{O}(g)$ means that $f$ is bounded above up to a constant factor by $g$, asymptotically.

### 3.2 Problem Formulation

We consider a connected network consisting of a set $\mathcal{V}$ of $K$ devices. Each device $i \in [K]$ has its data distribution $\mathcal{D}_i$ supported on domain $\Xi_i := (\mathcal{X}_i, \mathcal{Y}_i)$. The connectivity among devices is represented as an undirected connected communication graph $\mathcal{G}$ having the set $\mathcal{E} \subseteq \mathcal{V} \times \mathcal{V}$ of edges, as illustrated in Fig. 1. The set of neighbors of device $i$ is defined as $\mathcal{N}_i = \{j | (i, j) \in \mathcal{E}\}$ whose cardinality is $|\mathcal{N}_i| = d_i$. Note that $(i, j) \in \mathcal{E}$ if and only if devices $i$ and $j$ are connected by a communication link; in other words, these devices can exchange information directly. All devices collaborate to solve the optimization problem given by

$$\min_{\boldsymbol{\Theta} \in \mathbb{R}^d} \sum_{i=1}^{K} \frac{n_i}{n} f_i(\boldsymbol{\Theta}) \quad \text{where} \quad f_i(\boldsymbol{\Theta}) = \mathbb{E}_{\xi_i \sim \mathcal{D}_i}[\ell(\boldsymbol{\Theta}; \xi_i)], \tag{1}$$

where $\xi_i := (x_i, y_i)$ denotes the set of features $x_i$ and labels $y_i$ of device $i$. The function $\ell(\boldsymbol{\Theta}, \xi_i)$ is the cost of predicting $y_i$ from $x_i$, where $\boldsymbol{\Theta}$ denotes the model parameters, e.g., the weights/biases of a neural network. Here, $n_i$ denotes the number of training examples drawn from $\mathcal{D}_i$ and $n = \sum_{i=1}^{K} n_i$ is the total number of examples.

Without loss of generality, we assume in the remainder that all devices have the same number of samples, and therefore $n_i/n = 1/K$, $\forall i \in [K]$. In this case, problem (1) writes as

$$\min_{\boldsymbol{\Theta} \in \mathbb{R}^d} \frac{1}{K} \sum_{i=1}^{K} f_i(\boldsymbol{\Theta}). \tag{2}$$

One way to solve (1) in a decentralized way is to use vanilla decentralized SGD (DSGD) (Yuan et al., 2016). As shown in Algorithm 1, each device in DSGD performs two steps: (i) a local stochastic gradient

---

**Algorithm 1** VANILLA DECENTRALIZED SGD (DSGD)

---

1: **for** $t$ **in** $0, \ldots, T-1$ **do** *in parallel for all devices* $i \in [K]$
2:      Sample a mini-batch of size $\{\xi_j^t\}_{j=1}^B$, compute gradient $\boldsymbol{g}_i(\boldsymbol{\theta}_i^t) := \frac{1}{B} \sum_{j=1}^B \nabla \ell(\boldsymbol{\theta}_i^t, \xi_j^t)$
3:      $\boldsymbol{\theta}_i^{t+\frac{1}{2}} := \boldsymbol{\theta}_i^t - \eta \boldsymbol{g}_i(\boldsymbol{\theta}_i^t)$
4:      Send $\boldsymbol{\theta}_i^{t+\frac{1}{2}}$ to neighbors
5:      $\boldsymbol{\theta}_i^{t+1} := \sum_{j=1}^K W_{ij} \boldsymbol{\theta}_j^{t+\frac{1}{2}}$
6: **end for**

---

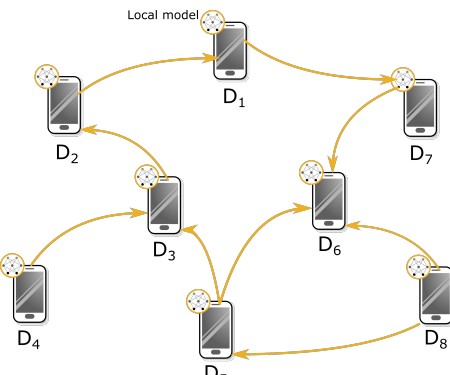

Figure 1: Illustration of a graph topology ($K = 8$) and the interactions between the devices ($D_1 - D_8$).

update (Line 3) using the learning rate $\eta$, and (ii) a consensus operation in which it averages its model with its neighbors' models (Lines 4-5) using the weights of the connectivity (mixing) matrix of the network $\boldsymbol{W} = [W_{ij}] \in \mathbb{R}^{K \times K}$. The mixing matrix $\boldsymbol{W}$ is often assumed to be a symmetric ($\boldsymbol{W} = \boldsymbol{W}^T$) and doubly stochastic ($\boldsymbol{W1} = \mathbf{1}, \mathbf{1}^T\boldsymbol{W} = \mathbf{1}^T$) matrix, such that $W_{ij} \in [0, 1]$, and if $(i, j) \notin \mathcal{E}$, then $W_{ij} = 0$. Assuming $\boldsymbol{W}$ to be symmetric and doubly stochastic is crucial to ensure that the devices achieve consensus in terms of converging to the same stationary point.

While formulating problem (2), we assume that the target distribution is given by

$$\overline{\mathcal{D}} = \frac{1}{K}\sum_{i=1}^{K}\mathcal{D}_i. \tag{3}$$

However, the heterogeneity of local data owned by the devices involved in the learning presents a significant challenge in the FL setting. In fact, models resulting from solving (2) lack robustness to distribution shifts and are vulnerable to adversarial attacks (Bhagoji et al., 2019). This is mainly due to the fact that the target distribution may be significantly different from $\overline{\mathcal{D}}$ in practice.

## 4 Proposed Solution

Our aim is to learn a global model $\boldsymbol{\Theta}$ from heterogeneous data coming from these possibly non-identical data distributions of the devices in a decentralized manner. To account for heterogeneous data distribution across devices, the authors in (Mohri et al., 2019) proposed agnostic FL, where the target distribution is given by

$$\mathcal{D}_{\boldsymbol{\lambda}} = \sum_{i=1}^{K}\lambda_i\mathcal{D}_i, \tag{4}$$

where the weighting vector $\boldsymbol{\lambda}$ belongs to the $K$-dimensional simplex, $\Delta = \{\boldsymbol{\lambda} = (\lambda_1, \dots, \lambda_K)^T \in \mathbb{R}_+^K : \sum_{i=1}^{K}\lambda_i = 1\}$. Note that this target distribution is more general than $\overline{\mathcal{D}}$ and it reduces to $\overline{\mathcal{D}}$ when $\lambda_i = 1/K, \forall i \in [K]$.

Unlike $\overline{\mathcal{D}}$ which gives equal weight to all distributions $\{\mathcal{D}_i\}_{i=1}^K$ during training, $\mathcal{D}_{\boldsymbol{\lambda}}$ is rather a mixture of devices' distributions, where the unknown mixture weight $\boldsymbol{\lambda}$ is learned during training and not known a priori. In this case, the distributionally robust empirical loss problem is given by the following min-max optimization problem

$$\min_{\boldsymbol{\Theta} \in \mathbb{R}^d} \max_{\boldsymbol{\lambda} \in \Delta} \sum_{i=1}^{K}\lambda_i f_i(\boldsymbol{\Theta}). \tag{5}$$

---

**Algorithm 2** DISTRIBUTIONALLY ROBUST DECENTRALIZED SGD (DR-DSGD)

---

1: **for** $t$ **in** $0, \ldots, T-1$ **do** *in parallel for all devices* $i \in [K]$
2:     Sample a mini-batch of size $\{\xi_j^t\}_{j=1}^B$, compute the gradient $\boldsymbol{g}_i(\boldsymbol{\theta}_i^t) := \frac{1}{B} \sum_{j=1}^B \nabla \ell(\boldsymbol{\theta}_i^t, \xi_j^t)$ and
      $h(\boldsymbol{\theta}_i^t; \mu) := \exp\left(\frac{1}{B} \sum_{j=1}^B \ell(\boldsymbol{\theta}_i^t, \xi_j^t)/\mu\right)$
3:     $\boldsymbol{\theta}_i^{t+\frac{1}{2}} := \boldsymbol{\theta}_i^t - \eta \times \frac{h_i(\boldsymbol{\theta}_i^t; \mu)}{\mu} \times \boldsymbol{g}_i(\boldsymbol{\theta}_i^t)$
4:     Send $\boldsymbol{\theta}_i^{t+\frac{1}{2}}$ to neighbors
5:     $\boldsymbol{\theta}_i^{t+1} := \sum_{j=1}^K W_{ij} \boldsymbol{\theta}_j^{t+\frac{1}{2}}$
6: **end for**

---

Although several distributed algorithms (Mohri et al., 2019; Reisizadeh et al., 2020; Deng et al., 2020; Hamer et al., 2020) have been proposed for (5), solving it in a decentralized fashion (in the absence of a PS) is a challenging task. Interestingly, when introducing a regularization term in (5) and by appropriately choosing the regularization function, the min-max optimization problem can be reduced to a robust minimization problem that can be solved in a decentralized manner, as shown later on. Specifically, the regularized version of problem (5) can be written as follows

$$\min_{\boldsymbol{\Theta} \in \mathbb{R}^d} \max_{\boldsymbol{\lambda} \in \Delta} \sum_{i=1}^K \lambda_i f_i(\boldsymbol{\Theta}) - \mu \phi(\boldsymbol{\lambda}, 1/K), \tag{6}$$

where $\mu > 0$ is a regularization parameter, and $\phi(\boldsymbol{\lambda}, 1/K)$ is a divergence measure between $\{\lambda_i\}_{i=1}^K$ and the uniform probability that assigns the same weight $1/K$ to every device's distribution. The function $\phi$ can be seen as a penalty that ensures that the weight $\lambda_i$ is not far away from $1/K$. Note that $\mu \to 0$ gives the distributionally robust problem (5) while when $\mu \to \infty$, we recover the standard empirical risk minimization problem (2).

Although different choices of the $\phi$-divergence can be considered in (6), the robust optimization community has been particularly interested in the Kullback–Leibler (KL) divergence owing to its simplified formulation (Esfahani & Kuhn, 2018). In fact, when we consider the KL divergence, i.e. $\phi(\boldsymbol{\lambda}, 1/K) = \sum_{i=1}^K \lambda_i \log(\lambda_i K)$, then by exactly maximizing over $\boldsymbol{\lambda} \in \Delta$, the min-max problem, given in (6), is shown to be equivalent to (Huang et al., 2021, Lemma 1)

$$\min_{\boldsymbol{\Theta} \in \mathbb{R}^d} \mu \log\left(\frac{1}{K} \sum_{i=1}^K \exp\left(f_i(\boldsymbol{\Theta})/\mu\right)\right). \tag{7}$$

Since $\log(\cdot)$ is a monotonically increasing function, then instead of solving (7), we simply solve the following problem

$$\min_{\boldsymbol{\Theta} \in \mathbb{R}^d} F(\boldsymbol{\Theta}) \triangleq \frac{1}{K} \sum_{i=1}^K F_i(\boldsymbol{\Theta}), \tag{8}$$

where $F_i(\boldsymbol{\Theta}) = \exp\left(f_i(\boldsymbol{\Theta})/\mu\right), \forall i \in [K]$.

**Remark 1.** *Note that problem (6) is equivalent to problem (5) as $\mu \to 0$. The proximity of the solutions of (6) to (5) in the **convex** case is discussed in Appendix A.2. In the remainder of this work, we focus on solving the regularized distributional robust optimization problem (Mohri et al., 2019; Deng et al., 2020) given in (6).*

Although any decentralized learning algorithm can be used to solve (8), the focus of this paper is to propose a distributionally robust implementation of DSGD. Our framework, coined Distributionally Robust Decentralized SGD (DR-DSGD), follows similar steps as DSGD with the main difference in the local update step

$$\boldsymbol{\theta}_i^{t+1} = \sum_{i=1}^K W_{ij} \left(\boldsymbol{\theta}_i^t - \frac{\eta}{\mu} h(\boldsymbol{\theta}_i^t; \mu) \boldsymbol{g}_i(\boldsymbol{\theta}_i^t)\right), \tag{9}$$

where $h(\boldsymbol{\theta}_i^t;\mu) := \exp\left(\frac{1}{B}\sum_{j=1}^B \ell(\boldsymbol{\theta}_i^t,\xi_j^t)/\mu\right)$ and $B$ is the mini-batch size $\{\xi_j^t\}_{j=1}^B$. Introducing the term $h(\boldsymbol{\theta}_i^t;\mu)/\mu$ in line 3 of Algorithm 2 makes the algorithm more robust to the heterogeneous setting and ensures fairness across devices, as will be shown in the numerical simulations section.

## 5   Convergence Analysis

This section provides a theoretical analysis of the convergence rate of the DR-DSGD algorithm. Before stating the main results of the paper, we make the following assumptions.

**Assumption 1. (Smoothness)** There exist constants $L_F$ and $L_1$, such that $\forall \boldsymbol{\theta_1}, \boldsymbol{\theta_2} \in \mathbb{R}^d$, we have

$$\|\nabla F(\boldsymbol{\theta_1}) - \nabla F(\boldsymbol{\theta_2})\| \le L_F \|\boldsymbol{\theta_1} - \boldsymbol{\theta_2}\|, \tag{10}$$

$$\mathbb{E}[\|\boldsymbol{g}_i(\boldsymbol{\theta_1}) - \boldsymbol{g}_i(\boldsymbol{\theta_2})\|] \le L_1 \|\boldsymbol{\theta_1} - \boldsymbol{\theta_2}\|. \tag{11}$$

**Assumption 2. (Gradient Boundedness)** The gradient of $f_i(\cdot)$ is bounded, i.e. there exits $G_1$ such that $\forall \boldsymbol{\theta} \in \mathbb{R}^d$, we have

$$\|\nabla f_i(\boldsymbol{\theta})\| \le G_1. \tag{12}$$

**Assumption 3. (Variance Boundedness)** The variances of stochastic gradient $g_i(\cdot)$ and the function $l_i(\cdot,\xi)$ are bounded, i.e. there exit positive scalars $\sigma_1$ and $\sigma_2$ such that $\forall \boldsymbol{\theta} \in \mathbb{R}^d$, we have

$$\mathbb{E}\left[|\ell(\boldsymbol{\theta},\xi_i) - f_i(\boldsymbol{\theta})|^2\right] \le \sigma_1^2, \tag{13}$$

$$\mathbb{E}\left[\|\boldsymbol{g}_i(\boldsymbol{\theta}) - \nabla f_i(\boldsymbol{\theta})\|^2\right] \le \sigma_2^2. \tag{14}$$

**Assumption 4. (Function Boundedness)** The function $F(\cdot)$ is lower bounded, i.e. $F_{inf} = \inf_{\boldsymbol{\theta}\in\mathbb{R}^d} F(\boldsymbol{\theta})$ such that $F_{inf} > -\infty$ and the functions $\ell(\cdot,\xi_i)$ are bounded, i.e. there exists $M > 0$ such that $\forall \boldsymbol{\theta} \in \mathbb{R}^d$, we have

$$\mathbb{E}\left[|\ell(\theta,\xi_i)|\right] \le M. \tag{15}$$

**Assumption 5. (Spectral Norm)** The spectral norm, defined as $\rho = \|\mathbb{E}\left[\boldsymbol{W}^T\boldsymbol{W}\right] - \boldsymbol{J}\|$, is assumed to be less than 1.

Assumptions 1-5 are key assumptions that are often used in the context of distributed and compositional optimization (Wang et al., 2016; 2017; Li et al., 2020e; Huang et al., 2021). Note that (15) can be fulfilled by imposing loss clipping (Xu et al., 2006; Wu & Liu, 2007; Yang et al., 2010) to most commonly-used loss functions. Furthermore, although the categorical cross-entropy function is not bounded upwards, it will only take on large values if the predictions are wrong. In fact, under some mild assumptions, the categorical cross-entropy will be around the entropy of a $M$-class uniform distribution, which is $\log(M)$ (refer to Appendix A.1 for a more in-depth discussion). Considering a relatively moderate number of classes, e.g., $M = 100$, we get $\log(M) < 5$. Thus, the cross-entropy will generally be relatively small in practice, and the assumption will hold in practice.

**Remark 2.** *It is worth mentioning that since the exponential function is convex, and non-decreasing function, then $\{\exp(f_i(\cdot))\}_{i=1}^K$ is convex when $\{f_i(\cdot)\}_{i=1}^K$ is convex. In this case, the convergence of the proposed method follows directly from existing results in the literature (Yuan et al., 2016).*

In the remainder of this section, we focus on the non-convex setting and we start by introducing the following matrices

$$\boldsymbol{\theta}^t = \left[\boldsymbol{\theta}_1^t, \ldots, \boldsymbol{\theta}_K^t\right], \tag{16}$$

$$\nabla \boldsymbol{F}^t = \left[\nabla \boldsymbol{F}_1(\boldsymbol{\theta}_1^t), \ldots, \nabla \boldsymbol{F}_K(\boldsymbol{\theta}_K^t)\right], \tag{17}$$

$$\boldsymbol{U}^t = \frac{1}{\mu}\left[h(\boldsymbol{\theta}_1^t;\mu)\boldsymbol{g}_1(\boldsymbol{\theta}_1^t), \ldots, h(\boldsymbol{\theta}_K^t;\mu)\boldsymbol{g}_K(\boldsymbol{\theta}_K^t)\right]. \tag{18}$$

**Remark 3.** *In the non-convex case, the convergence analysis is more challenging than in the case of DSGD. In fact, due to the compositional nature of the local loss functions, the stochastic gradients are biased, i.e.*

$$\mathbb{E}\left[\exp\left(\ell(\boldsymbol{\theta}_i^t, \xi_i^t)/\mu\right)\boldsymbol{g}_i(\boldsymbol{\theta}_i^t)\right] \neq \exp\left(f_i(\boldsymbol{\theta}_i^t)/\mu\right)\nabla f_i(\boldsymbol{\theta}_i^t). \tag{19}$$

To proceed with the analysis, we start by writing the matrix form of the update rule (9) as

$$\boldsymbol{\theta}^{t+1} = \left(\boldsymbol{\theta}^t - \eta\boldsymbol{U}^t\right)\boldsymbol{W}. \tag{20}$$

Multiplying both sides of the update rule (20) by $\mathbf{1}/K$, we get

$$\bar{\boldsymbol{\theta}}^{t+1} = \bar{\boldsymbol{\theta}}^t - \frac{\eta}{K}\boldsymbol{U}^t\mathbf{1}, \tag{21}$$

where $\bar{\boldsymbol{\theta}}^t$ is the averaged iterate across devices defined as $\bar{\boldsymbol{\theta}}^t = \frac{1}{K}\sum_{i=1}^{K}\boldsymbol{\theta}_i^t$. Now, we are in a position to introduce our first Lemma.

**Lemma 1.** *For any matrix $\boldsymbol{A} \in \mathbb{R}^{d \times K}$, we have*

$$\mathbb{E}\left[\|\boldsymbol{A}\left(\boldsymbol{W}^n - \boldsymbol{J}\right)\|_F^2\right] \leq \rho^n\|\boldsymbol{A}\|_F^2. \tag{22}$$

*Proof.* The details of the proof can be found in Appendix A.4. $\square$

The following lemma provides some of the inequalities that we need for the proof of the main result.

**Lemma 2.** *From Assumption 4, we have that*

(a) *There exists a constant $L_2(\mu)$ such that $\forall y_1, y_2 \in \mathcal{Y}$, we have*

$$\mathbb{E}[|\exp(y_1) - \exp(y_2)|] \leq L_2(\mu)|y_1 - y_2|, \tag{23}$$

*where $\mathcal{Y} \triangleq \{y = \ell(\boldsymbol{\theta}, \xi_i)/\mu \text{ such that } \boldsymbol{\theta} \in \mathbb{R}^d\}$ is the range of functions $\{\ell(\boldsymbol{\theta}, \xi_i)/\mu\}$.*

(b) *There exists a constant $G_2(\mu)$ such that $\forall \theta \in \mathbb{R}^d$, we have*

$$\mathbb{E}[|\exp(\ell(\boldsymbol{\theta}, \xi_i)/\mu)|] \leq G_2(\mu). \tag{24}$$

(c) *There exists a constant $\sigma_3(\mu)$ such that $\forall \theta \in \mathbb{R}^d$, we have*

$$\mathbb{E}\left[|\exp(\ell(\boldsymbol{\theta}, \xi_i)/\mu) - \exp(f_i(\boldsymbol{\theta})/\mu)|^2\right] \leq (\sigma_3(\mu))^2. \tag{25}$$

*Proof.* The details of the proof can be found in Appendix A.5. $\square$

Using Lemmas 1 and 2, we can now state Lemma 3, which gives an upper bound on the discrepancies among the local models.

**Lemma 3.** *Let $\eta$ satisfy $\eta L_\mu < \frac{\mu(1-\sqrt{\rho})}{4G_\mu\sqrt{\rho}}$. Provided that all local models are initiated at the same point, the discrepancies among the local models $\mathbb{E}\left[\|\boldsymbol{\theta}^t(\boldsymbol{I}-\boldsymbol{J})\|_F^2\right]$ can be upper bounded by*

$$\frac{1}{KT}\sum_{t=1}^{T}\mathbb{E}\left[\|\boldsymbol{\theta}^t(\boldsymbol{I}-\boldsymbol{J})\|_F^2\right] \leq \frac{2\eta^2\rho G_\mu^2[8\sigma_\mu^2(L_\mu^2+1)+G_\mu^2]}{\mu^2(1-\gamma_\mu)(1-\sqrt{\rho})^2}, \tag{26}$$

*where $\sigma_\mu = \max\{\sigma_1, \sigma_2, \sigma_3(\mu)\}$, $G_\mu = \max\{G_1, G_2(\mu)\}$, $L_\mu = \max\{L_F, L_1, L_2(\mu)\}$, and $\gamma_\mu = \frac{16\eta^2\rho L_\mu^2 G_\mu^2}{\mu^2(1-\sqrt{\rho})^2}$.*

*Proof.* The proof is deferred to Appendix A.6. In a nutshell, the proof uses the update rule of DR-DSGD, given in (20), the special property of the mixing matrix, and Lemma 1. $\square$

Next, we present the main theorem that states the convergence of our proposed algorithm.

**Theorem 1.** *Let $\eta$ satisfy $\eta L_\mu < \min\{\frac{\mu(1-\sqrt{\rho})}{8G_\mu\sqrt{\rho}}, 1\}$ and provided that all local models are initiated at the same point, then the averaged gradient norm is upper bounded as follows*

$$\frac{1}{T}\sum_{t=1}^{T}\mathbb{E}\left[\|\nabla F(\bar{\boldsymbol{\theta}}^t)\|^2\right] \leq \frac{2(F(\bar{\boldsymbol{\theta}}^1) - F_{inf})}{\eta T} + \frac{2G_\mu^2\sigma_\mu^2(L_\mu^2 + \mu^2)}{\mu^4 B} + \frac{16\rho\eta^2 L_\mu^2 G_\mu^2(G_\mu^2 + \mu^2)[8\sigma_\mu^2(L_\mu^2 + 1) + G_\mu^2]}{3\mu^8(1 - \sqrt{\rho})^2}.$$
(27)

*Proof.* The proof of Theorem 1 is detailed in Appendix A.7. $\qquad\square$

The first term of the right-hand side of (27) is neither affected by the graph parameters nor by the regularization parameter $\mu$ and it exhibits a linear speedup in $T$. This term exists also in the convergence analysis of DSGD (Wang et al., 2019; Lian et al., 2017). The graph topology affects the third term via the value of the spectral norm $\rho$. In fact, a smaller value of $\rho$ makes the third term smaller. Finally, the regularization parameter $\mu$ has an impact on both the second and third terms. A weakness of the derived bound is that as $\mu$ gets smaller, both the second and third terms grow larger. In the limit case, when $\mu \to 0$, both terms grow to infinity. In the next corollary, we limit ourselves to the case when $\mu \geq 1$ and show that under this setting DR-DSGD is guaranteed to converge.

**Remark 4.** *Our analysis is still valid in the case where the mixing matrix changes at every iteration. Similar theoretical guarantees hold provided that the matrices $\{\boldsymbol{W}^t\}_{t=1}^{T}$ are independent and identically distributed and their spectral norm $\rho^t < 1$.*

When $\mu \geq 1$, it can be seen from the proof of Lemma 2, that we can find constants, $L_2$, $G_2$, and $\sigma_3$, that do not depend on $\mu$. In this case, (27) can be written as

$$\frac{1}{T}\sum_{t=1}^{T}\mathbb{E}\left[\|\nabla F(\bar{\boldsymbol{\theta}}^t)\|^2\right] \leq \frac{2(F(\bar{\boldsymbol{\theta}}^1) - F_{inf})}{\eta T} + \frac{2G^2\sigma^2(L^2 + 1)}{B} + \frac{16\rho\eta^2 L^2 G^2(G^2 + 1)[8\sigma^2(L^2 + 1) + G_2]}{3(1 - \sqrt{\rho})^2}.$$
(28)

Furthermore, if the learning rate $\eta$ and the mini-batch size $B$ are chosen properly, we obtain the following corollary.

**Corollary 1.** *If $\mu \geq 1$ and if we choose $\eta = \frac{1}{2L + \sqrt{T/K}}$ and $B = \sqrt{KT}$, then we have*

$$\frac{1}{T}\sum_{t=1}^{T}\mathbb{E}\left[\|\nabla F(\bar{\boldsymbol{\theta}}^t)\|^2\right] = \mathcal{O}\left(\frac{1}{\sqrt{KT}} + \frac{K}{T}\right).$$
(29)

We show empirically in the next section that considering a regularization parameter such that $\mu \geq 1$ still leads to significant gains in terms of the worst distribution test accuracy and improves fairness across devices compared to DSGD.

## 6 Experiments

In this section, we validate our theoretical results and show the communication-efficiency, robustness and fairness of our proposed approach, DR-DSGD, compared to its non-robust counterpart DSGD.

### 6.1 Simulation Settings

For our experiments, we consider the image classification task using two main datasets: FASHION MNIST (Xiao et al., 2017) and CIFAR10 (Krizhevsky et al., 2009). We implement DR-DSGD and DSGD algorithms using PyTorch. For FASHION MNIST, we use an MLP model with ReLU activations having two hidden layers with 128 and 64 neurons, respectively. For the CIFAR10 dataset, we use a CNN model composed of

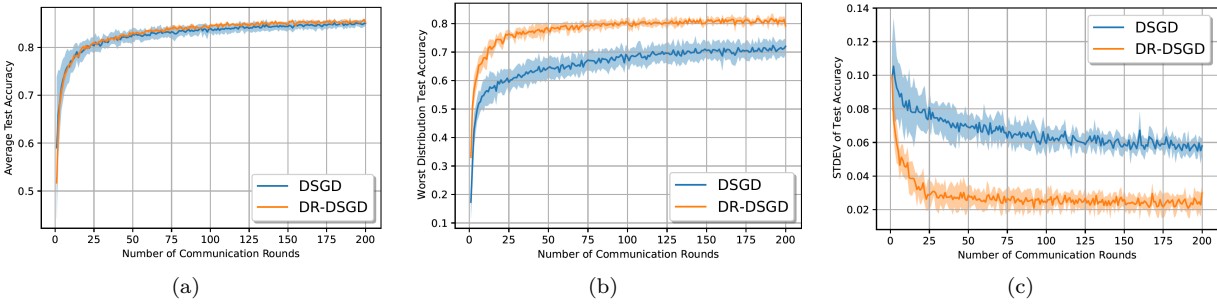

(a)              (b)              (c)

Figure 2: Performance comparison between DR-DSGD and DSGD in terms of: (a) average test accuracy, (b) worst test accuracy, and (c) STDEV of test accuracy for FASHION MNIST dataset.

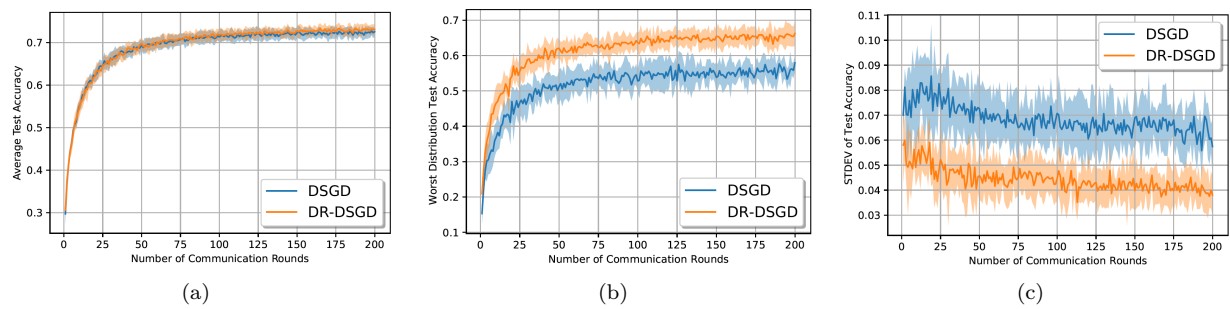

(a)              (b)              (c)

Figure 3: Performance comparison between DR-DSGD and DSGD in terms of: (a) average test accuracy, (b) worst test accuracy, and (c) STDEV of test accuracy for CIFAR10 dataset.

three convolutional layers followed by two fully connected layers, each having 500 neurons. For each dataset, we distribute the data across the $K$ devices in a pathological non-IID way, as in (McMahan et al., 2017), to mimic an actual decentralized learning setup. More specifically, we first order the samples according to the labels and divide data into shards of equal sizes. Finally, we assign each device the same number of chunks. This will ensure a pathological non-IID partitioning of the data, as most devices will only have access to certain classes and not all of them. Unless explicitly stated, we choose the learning rate $\eta = \sqrt{K/T}$ and the mini-batch size $B = \sqrt{KT}$.

For the graph generation, we generate randomly a network consisting of $K$ devices with a connectivity ratio $p$ using the networkx package (Hagberg et al., 2008). The parameter $p$ measures the sparsity of the graph. While smaller values of $p$ lead to a sparser graph, the generated graph becomes denser as $p$ approaches 1. We use the Metropolis weights to construct the mixing matrix $\boldsymbol{W}$ as follow

$$W_{ij} = \begin{cases} 1/\left(1 + \max\{d_i, d_j\}\right), & \text{if } (j,i) \in \mathcal{E}, \\ 0, & \text{if } (j,i) \notin \mathcal{E} \text{ and } j \neq i, \\ 1 - \sum_{l \in \mathcal{N}_i} W_{il}, & \text{if } j = i, \end{cases}$$

Unless otherwise stated, the graphs used in our experiments are of the Erdős-Rényi type.

## 6.2 Robustness & Communication-Efficiency

In this section, we consider $K = 10$ devices and $\mu = 6$. For FASHION MNIST, we consider a value of $p = 0.3$ while we take $p = 0.5$ for CIFAR10. For each experiment, we report both the average test accuracy, the worst distribution test accuracy, and their corresponding one standard error shaded area based on five runs. The worst distribution test accuracy is defined as the worst of all test accuracies. The performance comparison between DR-DSGD and DSGD for FASHION MNIST and CIFAR10 dataset is reported in Figs. 2 and 3, respectively. Both experiments show that while DR-DSGD achieves almost the same average test

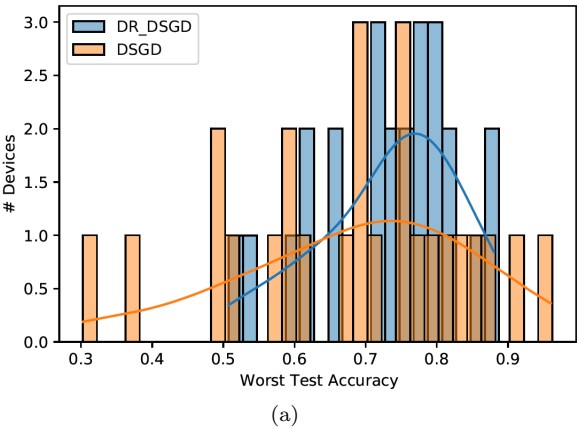 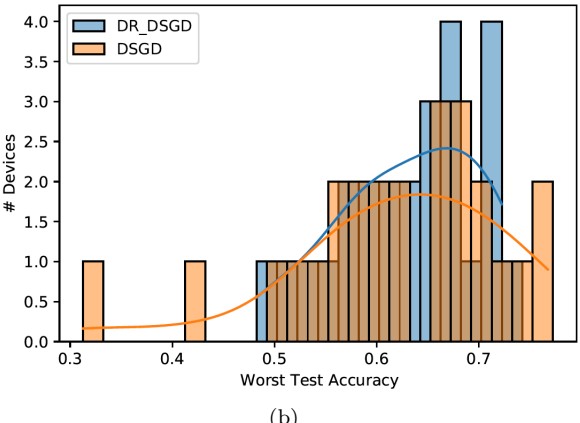

Figure 4: Performance comparison between DR-DSGD and DSGD in terms of worst test accuracy distribution for: (a) FASHION MNIST and (b) CIFAR10 datasets.

accuracy as DSGD, it significantly outperforms DSGD in terms of the worst distribution test accuracy. For the gap between both algorithms, the improvement, in terms of the worst distribution test accuracy, is of the order of 7% for FASHION MNIST and 10% for CIFAR10 datasets, respectively. This is mainly due to the fact that while the ERM objective sacrifices the worst-case performance for the average one, DRO aims to lower the variance while maintaining good average performance across the different devices. Not only our proposed algorithm achieves better performance than DSGD, but it is also more communication-efficient. In fact, for the same metric requirement, DR-DSGD requires fewer communication rounds than DSGD. Since our approach exponentially increases the weight of high training loss devices, it converges much faster than DSGD. For instance, in the experiment using the FASHION MNIST dataset, DR-DSGD requires $10\times$ fewer iterations than DSGD to achieve 70% worst distribution test accuracy. Finally, in each experiment, we plot the standard deviation (STDEV) of the different devices' test accuracies for both algorithms. We can see from both Figs. 2(c) and 3(c) that DR-DSGD has a smaller STDEV compared to DSGD, which reflects that DR-DSGD promotes more fairness among the devices.

### 6.3 Fairness

From this section on, we consider $K = 25$. To investigate the fairness of the performance across the devices, we run the experiments on FASHION MNIST and CIFAR10 datasets reporting the final test accuracy on each device in the case when $\mu = 9$. In Figs. 4(a) and 4(b), we plot the worst test accuracy distribution across devices. We note that DR-DSGD results in a more concentrated distribution in both experiments, hence a fairer test accuracy distribution with lower variance. For instance, DR-DSGD reduces the variance of accuracies across all devices by 60% on average for the FASHION MNIST experiment while keeping almost the same average accuracy.

### 6.4 Tradeoff Between Fairness & Average Test Accuracy

In this section, we show how $\mu$ controls the trade-off between fairness and average test accuracy. To this end, we report, in Table 1, the average, and worst 10% test accuracy, as well as the STDEV based on five runs for $T = 300$ and for different values of $\mu$ for both datasets. As expected, higher values of $\mu$ give more weight to the regularization term; hence driving the values of $\lambda_i$ closer to the average weight $1/K$. Therefore, as $\mu$ increases, the average test accuracy increases but the worst (10%) test accuracy decreases. Conversely, the worst test accuracy increases as the value of $\mu$ decreases at the cost of a drop in the average test accuracy. Furthermore, the STDEV decreases for smaller values of $\mu$ ensuring a fairer performance across devices.

Table 1: Statistics of the test accuracy distribution for different values of $\mu$.

| **Dataset** | $\mu$ | **Average** (%) | **Worst 10%** (%) | **STDEV** |
|---|---|---|---|---|
| FMNIST | $\mu = 2$ | $71.5 \pm 1.3$ | $\mathbf{49.1} \pm 2.4$ | $\mathbf{11.4} \pm 0.3$ |
| | $\mu = 3$ | $72.3 \pm 1.1$ | $48.8 \pm 3$ | $11.8 \pm 1.5$ |
| | $\mu = 5$ | $\mathbf{73.4} \pm 2.4$ | $44.5 \pm 4.4$ | $13.4 \pm 2.1$ |
| CIFAR10 | $\mu = 2$ | $57.2 \pm 2.4$ | $\mathbf{50.9} \pm 1.5$ | $\mathbf{10.3} \pm 0.6$ |
| | $\mu = 3$ | $59.83 \pm 1.6$ | $48.9 \pm 1.8$ | $10.9 \pm 0.4$ |
| | $\mu = 5$ | $\mathbf{61} \pm 1.4$ | $44.9 \pm 1.9$ | $11.2 \pm 1.3$ |

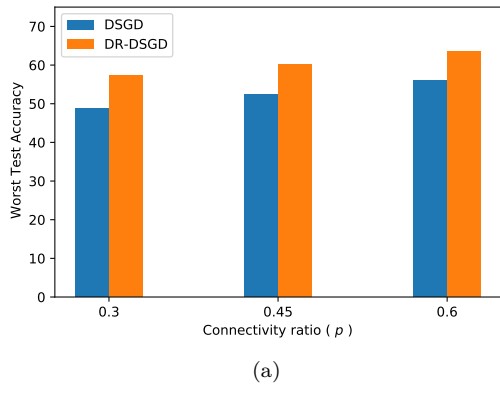

(a)

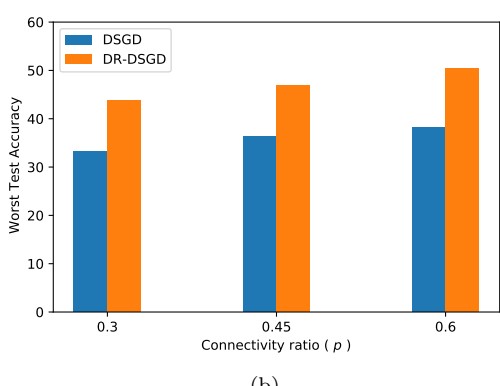

(b)

Figure 5: Performance comparison between DR-DSGD and DSGD in terms of worst test accuracy distribution for different values of $p$ for: (a) FASHION MNIST and (b) CIFAR10 datasets.

## 6.5 Impact of Graph Topology

In this section, we start by inspecting the effect of the graph sparsity on the performance of DR-DSGD and DSGD in terms of the worst distribution test accuracy by considering different connectivity ratios $p \in \{0.3, 0.45, 0.6\}$ and for $\mu = 6$. The results are reported in Fig 5 for both datasets for $\mu = 6$. We can see that as the graph becomes denser, i.e. as $p$ increases, the performance of both algorithms improves in terms of the worst test distribution. Nonetheless, it is clear that DR-DSGD outperforms DSGD for three different values of $p$ for both datasets. Next, we explore the performance of both algorithms on several other types of graph-types, specifically geometric (Fig. 6(a)), ring (Fig. 6(b)) and grid graphs (Fig. 6(c)). The first row represents the graph topology, while the second represents the worst distribution test accuracy as a function of the number of communication rounds for FASHION MNIST. We can see that DR-DSGD outperforms DSGD for the three graphs considered in Fig. 6 by achieving a higher worst distribution test accuracy. Furthermore, we note that both algorithms converge faster as the graph becomes denser. For instance, both algorithms require fewer communication rounds when the graph topology is geometric (Fig. 6(a)) compared to the ring graph (Fig. 6(b)).

## 6.6 Limitations

In this section, we highlight some of the limitations of our approach, while outlining several potential directions for future work.

- **Solving the unregularized distributionally robust problem:** Instead of solving (5), we propose DR-DSGD to solve its regularized form given in (5). This is mainly motivated by the fact that solving the min-max problem in (5) in a decentralized fashion is a challenging task. A weakness of

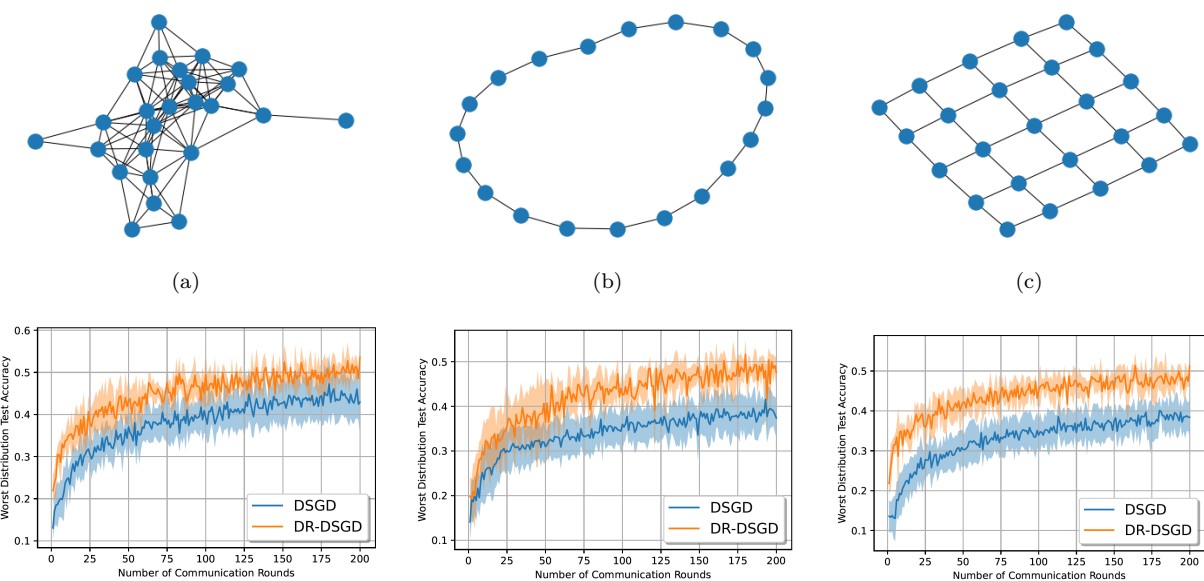

Figure 6: Performance comparison between DR-DSGD and DSGD in terms of worst test accuracy for: (a) geometric graph, (b) ring graph, and (c) grid graph for FASHION MNIST dataset.

the convergence analysis is that the derived convergence rate is obtained only for values of $\mu$ larger than one. For $\mu \to 0$, the bound, derived in Theorem 1, grows to infinity.

- **Relaxing some of the assumptions:** As pointed out in (Khaled et al., 2020), the bounded gradient assumption has been criticised in the FL literature. Adopting a more reasonable assumption is left for future work.

## 7 Conclusion

This paper proposes a distributionally robust decentralized algorithm, DR-DSGD, that builds upon the decentralized stochastic gradient descent (DSGD) algorithm. The proposed framework is the first to solve the distributionally robust learning problem over graphs. Simulation results indicate that our proposed algorithm is more robust across heterogeneous data distributions while being more communication-efficient than its non-robust counterpart, DSGD. Furthermore, the proposed approach ensures fairer performance across all devices compared to DSGD.

## Broader Impact Statement

This paper proposes a distributionally-robust decentralized learning algorithm. Our approach minimizes the maximum loss among the worst-case distributions across devices' data. As a result, even if the data distribution across devices is significantly heterogeneous, DR-DSGD guarantees a notion of fairness across devices. More specifically, the proposed algorithm ensures that the trained model has almost similar performance for all devices, rather than just performing well on a few of them while doing poorly on other devices.

## Acknowledgments

This work was supported in part by the Academy of Finland 6G Flagship under grant No. 318927, in part by project SMARTER, in part by projects EU-ICT IntellIoT under grant No. 957218, EUCHISTERA LearningEdge, CONNECT, Infotech-NOOR, and NEGEIN. We would like also to thank the anonymous reviewers for their insightful comments that led to improving the manuscript.

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

# A    Appendix

## A.1    Worst-case Bound on Categorical Cross-entropy

Let's first examine the behavior of a randomly initialized network. With random weights, the many units/layers will usually compound to result in the network outputting approximately uniform predictions. In a classification problem with $M$ classes, we will get probabilities of around $1/M$ for each category. In fact, the cross-entropy for a single data point is defined as

$$CE = -\sum_{m=1}^{M} y_m \log(\hat{y}_m), \tag{30}$$

where $y$ are the true probabilities (labels), $\hat{y}$ are the predictions. With hard labels (i.e. one-hot encoded), only a single $y_m$ is 1, and all others are 0. Thus, $CE$ reduces to $-\log(\hat{y}_m)$, where $m$ is now the correct class. With a randomly initialized network, we have $\hat{y}_m \sim 1/M$, therefore, we get $-\log(1/M) = \log(M)$. Since the training objective is usually to reduce cross-entropy, we can think of $\log(M)$ as a worst-case bound.

## A.2    Proximity of the Solutions of (6) to (5)

In this appendix, we discuss briefly the Proximity of the solutions of (6) to (5) in the **convex** case. According to (Qian et al., 2019), problem (6) can be re-written using the duality theory as

$$\min_{\Theta \in \mathbb{R}^d} \max_{\lambda \in \Delta} \sum_{i=1}^{K} \lambda_i f_i(\Theta) \tag{31}$$

$$s.t. \; \phi(\lambda, 1/K) \leq \tau \tag{32}$$

where to every $\mu$, we associate a $\tau$ value. From (Namkoong & Duchi, 2016; Duchi et al., 2016), for **convex** loss functions, we have that

$$\max_{\lambda \in \mathcal{P}_{\rho,K}} \sum_{i=1}^{K} \lambda_i \ell_i(\Theta) = \frac{1}{K} \sum_{i=1}^{K} \ell_i(\Theta) + \sqrt{\tau Var_{P_0}[\ell(\Theta, \xi)]} + o_{P_0}(K^{-\frac{1}{2}}), \tag{33}$$

where $\ell(\Theta) = [\ell_1(\Theta), \dots, \ell_K(\Theta)]^T \in \mathbb{R}^K$ is the vector of losses, $\mathcal{P}_0$ is the empirical probability distribution, and the ambiguity set $\mathcal{P}_{\rho,K}$ is defined as $\mathcal{P}_{\rho,K} = \{\lambda \in \mathbb{R}^K \sum_{i=1}^{K} \lambda_i = 1, \lambda \geq 0, \phi(\lambda, 1/K) \leq \tau\}$. Note that problem (5) is equivalent to (6) when $\tau = 0$. Thus, we can link the objective of the problem (6) to (5) as

$$\min_{\Theta \in \mathbb{R}^d} \max_{\lambda \in \mathcal{P}_{\rho,K}} \sum_{i=1}^{K} \lambda_i f_i(\Theta) = \min_{\Theta \in \mathbb{R}^d} \max_{\lambda \in \mathcal{P}_{0,K}} \sum_{i=1}^{K} \lambda_i f_i(\Theta) + \sqrt{\tau Var_{P_0}[\ell(\Theta, \xi)]}. \tag{34}$$

## A.3    Basic identities and inequalities

We start by summarizing the main identities and inequalities used in the proof. Let $\{\boldsymbol{a}_s\}_{s=1}^{S}$ be a sequence of vectors in $\mathbb{R}^d$, $b_1$ and $b_2$ two scalars, $\boldsymbol{C}_1$ and $\boldsymbol{C}_2$ two matrices, and $\epsilon > 0$, then we have

$$\left\| \sum_{s=1}^{S} \boldsymbol{a}_s \right\|^2 \leq S \sum_{s=1}^{S} \|\boldsymbol{a}_s\|^2. \tag{35}$$

$$2\langle \boldsymbol{a}_1, \boldsymbol{a}_2 \rangle = \|\boldsymbol{a}_1\|^2 + \|\boldsymbol{a}_2\|^2 - \|\boldsymbol{a}_1 - \boldsymbol{a}_2\|^2. \tag{36}$$

$$2b_1 b_2 \leq \frac{b_1^2}{\epsilon} + \epsilon b_2^2, \forall \epsilon > 0. \tag{37}$$

$$(\text{Cauchy-Schwarz}) \quad |\text{Tr}\{\boldsymbol{C}_1 \boldsymbol{C}_2\}| \leq \|\boldsymbol{C}_1\|_F \|\boldsymbol{C}_2\|_F. \tag{38}$$

### A.4 Proof of Lemma 1

Let $\boldsymbol{a}_i^T$ denote the i$^{th}$ row vector of matrix $\boldsymbol{A}$ and $\boldsymbol{e}_i$ the i$^{th}$ vector of the canonical basis of $\mathbb{R}^K$, then we can write

$$\begin{aligned}
&\mathbb{E}\left[\|\boldsymbol{A}\left(\boldsymbol{W}^n - \boldsymbol{J}\right)\|_F^2\right] \\
&= \sum_{i=1}^{K} \left\|\boldsymbol{a}_i^T \left(\boldsymbol{W}^n \boldsymbol{e}_i - \frac{\mathbf{1}}{K}\right)\right\|^2 \\
&\leq \sum_{i=1}^{K} \|\boldsymbol{a}_i^T\|^2 \left\|\boldsymbol{W}^n \boldsymbol{e}_i - \frac{\mathbf{1}}{K}\right\|^2.
\end{aligned} \tag{39}$$

From (Lian et al., 2017, Lemma 5), we have

$$\left\|\boldsymbol{W}^n \boldsymbol{e}_i - \frac{\mathbf{1}}{K}\right\|^2 \leq \rho^n. \tag{40}$$

Replacing equation 40 in equation 39, we get

$$\mathbb{E}\left[\|\boldsymbol{A}\left(\boldsymbol{W}^n - \boldsymbol{J}\right)\|_F^2\right] \leq \rho^n \|\boldsymbol{A}\|_F^2. \tag{41}$$

Hence, the proof is completed.

### A.5 Proof of Lemma 2

From Assumption 4, we have that the function $\theta \mapsto \ell(\theta, \xi)$ is bounded, then $\theta \mapsto \exp(\ell(\theta, \xi))$ is $C^1$ on a compact, and as a consequence, there exists $L_e > 0$ such that

$$\mathbb{E}[|\exp(z_1) - \exp(z_2)|] \leq L_e |z_1 - z_2|, \tag{42}$$

where $z_i = \ell(\theta_i, \xi)$. Next, let $z_1 > z_2$ and define $t_i = \exp(z_i)$, then we can write

$$\mathbb{E}[t_1^{\frac{1}{\mu}} - t_2^{\frac{1}{\mu}}] = \frac{1}{\mu}\mathbb{E}\left[\int_{t_2}^{t_1} x^{\frac{1}{\mu}-1} dx\right] \leq \frac{\exp\left((\frac{1}{\mu}-1)M\right)}{\mu}\mathbb{E}[t_1 - t_2] = \frac{\exp\left((\frac{1}{\mu}-1)M\right)}{\mu}\mathbb{E}\left[\exp(z_1) - \exp(z_2)\right]. \tag{43}$$

Using (42), we get

$$\mathbb{E}[t_1^{\frac{1}{\mu}} - t_2^{\frac{1}{\mu}}] \leq \exp\left((\frac{1}{\mu}-1)M\right) L_e \mathbb{E}\left[\frac{z_1}{\mu} - \frac{z_2}{\mu}\right] \tag{44}$$

A similar result can be obtained if we considered $z_1 < z_2$, hence, we can write

$$\mathbb{E}\left[|t_1^{\frac{1}{\mu}} - t_2^{\frac{1}{\mu}}|\right] \leq \exp\left((\frac{1}{\mu}-1)M\right) L_e \mathbb{E}\left[\left|\frac{z_1}{\mu} - \frac{z_2}{\mu}\right|\right]. \tag{45}$$

Using the definition of $\mathcal{Y}$, we can write

$$\mathbb{E}\left[|\exp(y_1) - \exp(y_2)|\right] \leq L_2(\mu)|y_1 - y_2|, \tag{46}$$

where $L_2(\mu) := \exp\left(\left(\frac{1}{\mu} - 1\right)M\right) L_e$ and hence the proof of statement $(a)$.

For statement $(b)$, using Hoeffding's lemma, we can write

$$\mathbb{E}\left[\exp(\ell(\theta, \xi_i)/\mu)\right] \leq \exp\left(\frac{\mathbb{E}\left[\ell(\theta, \xi_i)\right]}{\mu} + \frac{M^2}{2\mu^2}\right) \leq \exp\left(\frac{M}{\mu} + \frac{M^2}{2\mu^2}\right). \tag{47}$$

By choosing $G_2(\mu) := \exp\left(\frac{M}{\mu} + \frac{M^2}{2\mu^2}\right)$, we prove statement $(b)$.

Finally, for statement $(c)$, we can write

$$\mathbb{E}\left[|\exp(\ell(\boldsymbol{\theta}, \xi_i)/\mu) - \exp(f_i(\boldsymbol{\theta})/\mu)|^2\right] \leq 2\left(\mathbb{E}\left[\exp(2\ell(\boldsymbol{\theta}, \xi_i)/\mu)\right] + \exp(2f_i(\boldsymbol{\theta})/\mu)\right). \tag{48}$$

Using Assumption 4 and Hoeffding's lemma, we get

$$\mathbb{E}\left[|\exp(\ell(\boldsymbol{\theta}, \xi_i)/\mu) - \exp(f_i(\boldsymbol{\theta})/\mu)|^2\right] \leq (\sigma_3(\mu))^2, \tag{49}$$

where $(\sigma_3(\mu))^2 := 2\left(\exp\left(\frac{2M}{\mu} + \frac{M^2}{\mu^2}\right) + \exp\left(\frac{2M}{\mu}\right)\right).$

### A.6   Proof of Lemma 3

Using the update rule (20) and the identity $\boldsymbol{W}\boldsymbol{J} = \boldsymbol{J}\boldsymbol{W} = \boldsymbol{J}$, we can write

$$\boldsymbol{\theta}^t(\boldsymbol{I} - \boldsymbol{J}) = \left(\boldsymbol{\theta}^{t-1} - \eta\boldsymbol{U}^{t-1}\right)\boldsymbol{W}(\boldsymbol{I} - \boldsymbol{J}) = \boldsymbol{\theta}^{t-1}(\boldsymbol{I} - \boldsymbol{J})\boldsymbol{W} - \eta\boldsymbol{U}^{t-1}\boldsymbol{W}(\boldsymbol{I} - \boldsymbol{J}). \tag{50}$$

Writing (50) recursively, we get

$$\boldsymbol{\theta}^t(\boldsymbol{I} - \boldsymbol{J}) = \boldsymbol{\theta}^0(\boldsymbol{I} - \boldsymbol{J})\boldsymbol{W}^t - \eta\sum_{\tau=0}^{t-1}\boldsymbol{U}^\tau\left(\boldsymbol{W}^{t-\tau} - \boldsymbol{J}\right) = -\eta\sum_{\tau=0}^{t-1}\boldsymbol{U}^\tau\left(\boldsymbol{W}^{t-\tau} - \boldsymbol{J}\right), \tag{51}$$

where we used the fact that all local models are initiated at the same point, i.e. $\boldsymbol{\theta}^0(\boldsymbol{I} - \boldsymbol{J})\boldsymbol{W}^t = \boldsymbol{0}$.

Thus, we can write

$$\frac{1}{KT}\sum_{t=1}^{T}\mathbb{E}\left[\|\boldsymbol{\theta}^t(\boldsymbol{I} - \boldsymbol{J})\|_F^2\right]$$

$$= \frac{\eta^2}{KT}\sum_{t=1}^{T}\mathbb{E}\left[\left\|\sum_{\tau=0}^{t-1}\boldsymbol{U}^\tau\left(\boldsymbol{W}^{t-\tau} - \boldsymbol{J}\right)\right\|_F^2\right]$$

$$= \frac{\eta^2}{KT}\sum_{t=1}^{T}\mathbb{E}\left[\left\|\sum_{\tau=0}^{t-1}\left(\boldsymbol{U}^\tau - \nabla\boldsymbol{F}^\tau + \nabla\boldsymbol{F}^\tau\right)\left(\boldsymbol{W}^{t-\tau} - \boldsymbol{J}\right)\right\|_F^2\right]$$

$$\leq \frac{2\eta^2}{KT}\sum_{t=1}^{T}\mathbb{E}\left[\left\|\sum_{\tau=0}^{t-1}\left(\boldsymbol{U}^\tau - \nabla\boldsymbol{F}^\tau\right)\left(\boldsymbol{W}^{t-\tau} - \boldsymbol{J}\right)\right\|_F^2\right] + \frac{2\eta^2}{KT}\sum_{t=1}^{T}\mathbb{E}\left[\left\|\sum_{\tau=0}^{t-1}\nabla\boldsymbol{F}^\tau\left(\boldsymbol{W}^{t-\tau} - \boldsymbol{J}\right)\right\|_F^2\right], \tag{52}$$

where we used (35) (for $S = 2$) in the last inequality. Let $\boldsymbol{B}_{\tau,t} = \boldsymbol{W}^{t-\tau} - \boldsymbol{J}$. We start by examining the first term of (52) by writing

$$
\mathbb{E}\left[\left\|\sum_{\tau=0}^{t-1}\left(\boldsymbol{U}^{\tau} - \nabla\boldsymbol{F}^{\tau}\right)\boldsymbol{B}_{\tau,t}\right\|_F^2\right]
$$

$$
= \sum_{\tau=0}^{t-1}\mathbb{E}\left[\|\left(\boldsymbol{U}^{\tau} - \nabla\boldsymbol{F}^{\tau}\right)\boldsymbol{B}_{\tau,t}\|_F^2\right] + \sum_{\tau=0}^{t-1}\sum_{\substack{\tau'=0,\\\tau'\neq\tau}}^{t-1}\mathbb{E}\left[\mathrm{Tr}\{\boldsymbol{B}_{\tau,t}^T(\boldsymbol{U}^{\tau} - \nabla\boldsymbol{F}^{\tau})^T(\boldsymbol{U}^{\tau'} - \nabla\boldsymbol{F}^{\tau'})\boldsymbol{B}_{\tau',t}\}\right]
$$

$$
\leq \sum_{\tau=0}^{t-1}\mathbb{E}\left[\|\boldsymbol{U}^{\tau} - \nabla\boldsymbol{F}^{\tau}\|_F^2\,\|\boldsymbol{B}_{\tau,t}\|_F^2\right] + \sum_{\tau=0}^{t-1}\sum_{\substack{\tau'=0,\\\tau'\neq\tau}}^{t-1}\mathbb{E}\left[\|\left(\boldsymbol{U}^{\tau} - \nabla\boldsymbol{F}^{\tau}\right)\boldsymbol{B}_{\tau,t}\|_F\,\|\left(\boldsymbol{U}^{\tau'} - \nabla\boldsymbol{F}^{\tau'}\right)\boldsymbol{B}_{\tau',t}\|_F\right]
$$

$$
\leq \sum_{\tau=0}^{t-1}\rho^{t-\tau}\mathbb{E}\left[\|\boldsymbol{U}^{\tau} - \nabla\boldsymbol{F}^{\tau}\|_F^2\right] + \sum_{\tau=0}^{t-1}\sum_{\substack{\tau'=0,\\\tau'\neq\tau}}^{t-1}\left\{\frac{\rho^{t-\tau}}{2\epsilon}\mathbb{E}\left[\|\boldsymbol{U}^{\tau} - \nabla\boldsymbol{F}^{\tau}\|_F^2\right] + \frac{\epsilon\rho^{t-\tau'}}{2}\mathbb{E}\left[\left\|\boldsymbol{U}^{\tau'} - \nabla\boldsymbol{F}^{\tau'}\right\|_F^2\right]\right\}, \quad (53)
$$

where we have used Lemma 1 and inequalities (37) and (38). Setting $\epsilon = \rho^{\frac{\tau'-\tau}{2}}$, we can further write (53) as

$$
\mathbb{E}\left[\left\|\sum_{\tau=0}^{t-1}\left(\boldsymbol{U}^{\tau} - \nabla\boldsymbol{F}^{\tau}\right)\boldsymbol{B}_{\tau,t}\right\|_F^2\right]
$$

$$
\leq \sum_{\tau=0}^{t-1}\rho^{t-\tau}\mathbb{E}\left[\|\boldsymbol{U}^{\tau} - \nabla\boldsymbol{F}^{\tau}\|_F^2\right] + \sum_{\tau=0}^{t-1}\sum_{\substack{\tau'=0,\\\tau'\neq\tau}}^{t-1}\frac{\rho^{t-\frac{\tau+\tau'}{2}}}{2}\left(\mathbb{E}\left[\|\boldsymbol{U}^{\tau} - \nabla\boldsymbol{F}^{\tau}\|_F^2 + \left\|\boldsymbol{U}^{\tau'} - \nabla\boldsymbol{F}^{\tau'}\right\|_F^2\right]\right)
$$

$$
\leq \sum_{\tau=0}^{t-1}\rho^{t-\tau}\mathbb{E}\left[\|\boldsymbol{U}^{\tau} - \nabla\boldsymbol{F}^{\tau}\|_F^2\right] + \sum_{\tau=0}^{t-1}\sum_{\substack{\tau'=0,\\\tau'\neq\tau}}^{t-1}\rho^{t-\frac{\tau+\tau'}{2}}\mathbb{E}\left[\|\boldsymbol{U}^{\tau} - \nabla\boldsymbol{F}^{\tau}\|_F^2\right]
$$

$$
\leq \sum_{\tau=0}^{t-1}\rho^{t-\tau}\mathbb{E}\left[\|\boldsymbol{U}^{\tau} - \nabla\boldsymbol{F}^{\tau}\|_F^2\right] + \sum_{\tau=0}^{t-1}\rho^{\frac{t-\tau}{2}}\mathbb{E}\left[\|\boldsymbol{U}^{\tau} - \nabla\boldsymbol{F}^{\tau}\|_F^2\right]\sum_{\substack{\tau'=0,\\\tau'\neq\tau}}^{t-1}\rho^{\frac{t-\tau'}{2}}
$$

$$
= \sum_{\tau=0}^{t-1}\rho^{t-\tau}\mathbb{E}\left[\|\boldsymbol{U}^{\tau} - \nabla\boldsymbol{F}^{\tau}\|_F^2\right] + \sum_{\tau=0}^{t-1}\rho^{\frac{t-\tau}{2}}\mathbb{E}\left[\|\boldsymbol{U}^{\tau} - \nabla\boldsymbol{F}^{\tau}\|_F^2\right]\left(\sum_{\tau'=0}^{t-1}\rho^{\frac{t-\tau'}{2}} - \rho^{\frac{t-\tau}{2}}\right)
$$

$$
\leq \sum_{\tau=0}^{t-1}\rho^{\frac{t-\tau}{2}}\mathbb{E}\left[\|\boldsymbol{U}^{\tau} - \nabla\boldsymbol{F}^{\tau}\|_F^2\right]\sum_{\tau'=0}^{t-1}\rho^{\frac{t-\tau'}{2}}
$$

$$
\leq \frac{\sqrt{\rho}}{1 - \sqrt{\rho}}\sum_{\tau=0}^{t-1}\rho^{\frac{t-\tau}{2}}\mathbb{E}\left[\|\boldsymbol{U}^{\tau} - \nabla\boldsymbol{F}^{\tau}\|_F^2\right], \quad (54)
$$

where in the last inequality, we used $\sum_{\tau=0}^{t-1}\rho^{\frac{t-\tau}{2}} = \sqrt{\rho}^t + \sqrt{\rho}^{t-1} + \cdots + \sqrt{\rho} \leq \frac{\sqrt{\rho}}{1-\sqrt{\rho}}$. Now, let's focus on finding an upper bound for the term $\mathbb{E}\left[\|\boldsymbol{U}^{\tau} - \nabla\boldsymbol{F}^{\tau}\|_F^2\right]$. To this end, we start by writing

$$
\|\boldsymbol{U}^{\tau} - \nabla\boldsymbol{F}^{\tau}\|_F^2 = \frac{1}{\mu^2}\sum_{i=1}^{K}\left\|h(\boldsymbol{\theta}_i^{\tau};\mu)\boldsymbol{g}_i(\boldsymbol{\theta}_i^{\tau}) - \exp\left(\frac{f_i(\boldsymbol{\theta}_i^{\tau})}{\mu}\right)\nabla f_i(\boldsymbol{\theta}_i^{\tau})\right\|^2. \quad (55)
$$

Next, we can write the following

$$
h(\boldsymbol{\theta}_i^\tau; \mu)\boldsymbol{g}_i(\boldsymbol{\theta}_i^\tau) - \exp\left(\frac{f_i(\boldsymbol{\theta}_i^\tau)}{\mu}\right)\nabla f_i(\boldsymbol{\theta}_i^\tau)
$$
$$
= h(\boldsymbol{\theta}_i^\tau; \mu)\boldsymbol{g}_i(\boldsymbol{\theta}_i^\tau) - h(\boldsymbol{\theta}_i^\tau; \mu)\boldsymbol{g}_i(\bar{\boldsymbol{\theta}}^\tau) + h(\boldsymbol{\theta}_i^\tau; \mu)\boldsymbol{g}_i(\bar{\boldsymbol{\theta}}^\tau) - h(\boldsymbol{\theta}_i^\tau; \mu)\nabla f_i(\bar{\boldsymbol{\theta}}^\tau) + h(\boldsymbol{\theta}_i^\tau; \mu)\nabla f_i(\bar{\boldsymbol{\theta}}^\tau)
$$
$$
- \exp\left(\frac{f_i(\boldsymbol{\theta}_i^\tau)}{\mu}\right)\nabla f_i(\bar{\boldsymbol{\theta}}^\tau) + \exp\left(\frac{f_i(\boldsymbol{\theta}_i^\tau)}{\mu}\right)\nabla f_i(\bar{\boldsymbol{\theta}}^\tau) - \exp\left(\frac{f_i(\boldsymbol{\theta}_i^\tau)}{\mu}\right)\nabla f_i(\boldsymbol{\theta}_i^\tau). \tag{56}
$$

Using the decomposition (56) in (55) and taking the expected value while using the inequality (35) (for $S = 4$), we get

$$
\mathbb{E}\left[\|\boldsymbol{U}^\tau - \nabla \boldsymbol{F}^\tau\|_F^2\right]
$$
$$
\leq \frac{4}{\mu^2}\sum_{i=1}^K \mathbb{E}\left[\left\|h(\boldsymbol{\theta}_i^\tau; \mu)\left(\boldsymbol{g}_i(\boldsymbol{\theta}_i^\tau) - \boldsymbol{g}_i(\bar{\boldsymbol{\theta}}^\tau)\right)\right\|^2\right] + \frac{4}{\mu^2}\sum_{i=1}^K \mathbb{E}\left[\left\|h(\boldsymbol{\theta}_i^\tau; \mu)\left(\boldsymbol{g}_i(\bar{\boldsymbol{\theta}}^\tau) - \nabla f_i(\bar{\boldsymbol{\theta}}^i)\right)\right\|^2\right]
$$
$$
+ \frac{4}{\mu^2}\sum_{i=1}^K \mathbb{E}\left[\left\|\nabla f_i(\bar{\boldsymbol{\theta}}^\tau)\left(h(\boldsymbol{\theta}_i^\tau; \mu) - \exp\left(\frac{f_i(\boldsymbol{\theta}_i^\tau)}{\mu}\right)\right)\right\|^2\right]
$$
$$
+ \frac{4}{\mu^2}\sum_{i=1}^K \mathbb{E}\left[\left\|\exp\left(\frac{f_i(\boldsymbol{\theta}_i^\tau)}{\mu}\right)\left(\nabla f_i(\bar{\boldsymbol{\theta}}^\tau) - \nabla f_i(\boldsymbol{\theta}_i^\tau)\right)\right\|^2\right]
$$
$$
\leq \frac{4(G_2(\mu))^2}{\mu^2}\sum_{i=1}^K \mathbb{E}\left[\|\boldsymbol{g}_i(\boldsymbol{\theta}_i^\tau) - \boldsymbol{g}_i(\bar{\boldsymbol{\theta}}^\tau)\|^2\right] + \frac{4(G_2(\mu))^2}{\mu^2}\sum_{i=1}^K \mathbb{E}\left[\|\boldsymbol{g}_i(\bar{\boldsymbol{\theta}}^\tau) - \nabla f_i(\bar{\boldsymbol{\theta}}^\tau)\|^2\right]
$$
$$
+ \frac{4G_1^2}{\mu^2}\sum_{i=1}^K \mathbb{E}\left[\left|h(\boldsymbol{\theta}_i^\tau; \mu) - \exp\left(\frac{f_i(\boldsymbol{\theta}_i^\tau)}{\mu}\right)\right|^2\right] + \frac{4(G_2(\mu))^2}{\mu^2}\sum_{i=1}^K \mathbb{E}\left[\|\nabla f_i(\bar{\boldsymbol{\theta}}^\tau) - \nabla f_i(\boldsymbol{\theta}_i^\tau)\|^2\right]
$$
$$
\leq \frac{8G_\mu^2 L_\mu^2}{\mu^2}\sum_{i=1}^K \mathbb{E}\left[\|\bar{\boldsymbol{\theta}}^\tau - \boldsymbol{\theta}_i^\tau\|^2\right] + \frac{8G_\mu^2\sigma_\mu^2(L_\mu^2 + 1)K}{\mu^2 B}
$$
$$
\leq \frac{8G_\mu^2 L_\mu^2}{\mu^2}\mathbb{E}\left[\|\boldsymbol{\theta}^\tau(\boldsymbol{I} - \boldsymbol{J})\|_F^2\right] + \frac{8G_\mu^2\sigma_\mu^2(L_\mu^2 + 1)K}{\mu^2}, \tag{57}
$$

where we used Assumptions 1-3 and defined the constants $\sigma_\mu = \max\{\sigma_1, \sigma_2, \sigma_3(\mu)\}$, $G_\mu = \max\{G_1, G_2(\mu)\}$ and $L_\mu = \max\{L_F, L_1, L_2(\mu)\}$. Going back to (54), and using (57), we can write

$$
\frac{1}{KT}\sum_{t=1}^T \mathbb{E}\left[\left\|\sum_{\tau=0}^{t-1}(\boldsymbol{U}^\tau - \nabla \boldsymbol{F}^\tau)\boldsymbol{B}_{\tau,t}\right\|_F^2\right]
$$
$$
\leq \frac{8\sqrt{\rho}G_\mu^2 L_\mu^2}{\mu^2(1 - \sqrt{\rho})}\frac{1}{KT}\sum_{t=1}^T\sum_{\tau=0}^{t-1}\rho^{\frac{t-\tau}{2}}\mathbb{E}\left[\|\boldsymbol{\theta}^\tau(\boldsymbol{I} - \boldsymbol{J})\|_F^2\right] + \frac{8\sqrt{\rho}G_\mu^2\sigma_\mu^2(L_\mu^2 + 1)}{\mu^2(1 - \sqrt{\rho})T}\sum_{t=1}^T\sum_{\tau=0}^{t-1}\rho^{\frac{t-\tau}{2}}
$$
$$
\leq \frac{8\sqrt{\rho}G_\mu^2 L_\mu^2}{\mu^2(1 - \sqrt{\rho})}\frac{1}{KT}\sum_{t=1}^T \mathbb{E}\left[\|\boldsymbol{\theta}^t(\boldsymbol{I} - \boldsymbol{J})\|_F^2\right]\sum_{\tau=0}^{T-t}\rho^{\frac{\tau}{2}} + \frac{8\sqrt{\rho}G_\mu^2\sigma_\mu^2(L_\mu^2 + 1)}{\mu^2(1 - \sqrt{\rho})T}\sum_{t=1}^T\sum_{\tau=0}^{T-t}\rho^{\frac{\tau}{2}}
$$
$$
\leq \frac{8\rho G_\mu^2 L_\mu^2}{\mu^2(1 - \sqrt{\rho})^2}\frac{1}{KT}\sum_{t=1}^T \mathbb{E}\left[\|\boldsymbol{\theta}^t(\boldsymbol{I} - \boldsymbol{J})\|_F^2\right] + \frac{8\rho G_\mu^2\sigma_\mu^2(L_\mu^2 + 1)}{\mu^2(1 - \sqrt{\rho})^2}. \tag{58}
$$

Now, we focus on the second term of (52). Following similar steps as when bounding the first term of (52), we get

$$
\mathbb{E}\left[\left\|\sum_{\tau=0}^{t-1}\nabla \boldsymbol{F}^\tau \boldsymbol{B}_{\tau,t}\right\|_F^2\right] \leq \frac{\sqrt{\rho}}{1 - \sqrt{\rho}}\sum_{\tau=0}^{t-1}\rho^{\frac{t-\tau}{2}}\mathbb{E}\left[\|\nabla \boldsymbol{F}^\tau\|_F^2\right]. \tag{59}
$$

Next, we look for an upper bound for the term $\mathbb{E}\left[\|\nabla \boldsymbol{F}^\tau\|_F^2\right]$. To this end, we start by writing

$$
\begin{aligned}
&\mathbb{E}\left[\|\nabla \boldsymbol{F}^\tau\|_F^2\right] \\
&= \frac{1}{\mu^2} \sum_{i=1}^K \mathbb{E}\left[\left\|\exp\left(\frac{f_i(\boldsymbol{\theta}_i^\tau)}{\mu}\right) \nabla f_i(\boldsymbol{\theta}_i^\tau)\right\|^2\right] \\
&\leq \frac{(G_2(\mu))^2}{\mu^2} \mathbb{E}\left[\|\nabla f_i(\boldsymbol{\theta}_i^\tau)\|^2\right] \\
&\leq \frac{G1^2 (G_2(\mu))^2 K}{\mu^2} \\
&\leq \frac{G_\mu^4 K}{\mu^2}.
\end{aligned}
\tag{60}
$$

Therefore, we get

$$
\frac{1}{KT} \sum_{t=1}^T \mathbb{E}\left[\left\|\sum_{\tau=0}^{t-1} \nabla \boldsymbol{F}^\tau \boldsymbol{B}_{\tau,t}\right\|_F^2\right] \leq \frac{G_\mu^4 \rho}{\mu^2 (1-\sqrt{\rho})^2}.
\tag{61}
$$

Next, using (52), (58), and (61), we can write

$$
\frac{1}{KT} \sum_{t=1}^T \mathbb{E}\left[\|\boldsymbol{\theta}^t(\boldsymbol{I}-\boldsymbol{J})\|_F^2\right] \leq \frac{16\eta^2 \rho L_\mu^2 G_\mu^2}{\mu^2 (1-\sqrt{\rho})^2} \frac{1}{KT} \sum_{t=1}^T \mathbb{E}\left[\|\boldsymbol{\theta}^t(\boldsymbol{I}-\boldsymbol{J})\|_F^2\right] + \frac{2\eta^2 \rho G_\mu^2 [8\sigma_\mu^2(L_\mu^2+1)+G_\mu^2]}{\mu^2 (1-\sqrt{\rho})^2}.
\tag{62}
$$

Let $\gamma_\mu = \frac{16\eta^2 \rho L_\mu^2 G_\mu^2}{\mu^2(1-\sqrt{\rho})^2}$, then we obtain

$$
\frac{1}{KT} \sum_{t=1}^T \mathbb{E}\left[\|\boldsymbol{\theta}^t(\boldsymbol{I}-\boldsymbol{J})\|_F^2\right] \leq \frac{2\eta^2 \rho G_\mu^2 [8\sigma_\mu^2(L_\mu^2+1)+G_\mu^2]}{\mu^2 (1-\gamma_\mu)(1-\sqrt{\rho})^2},
\tag{63}
$$

which concludes the proof of Lemma 3.

## A.7 Proof of Theorem 1

Since the objective function $F(\cdot)$ has a Lipschitz gradient, we can write

$$
F(\bar{\boldsymbol{\theta}}^{t+1}) - F(\bar{\boldsymbol{\theta}}^t) \leq \langle \nabla F(\bar{\boldsymbol{\theta}}^t), \bar{\boldsymbol{\theta}}^{t+1} - \bar{\boldsymbol{\theta}}^t \rangle + \frac{L_\mu}{2} \|\bar{\boldsymbol{\theta}}^{t+1} - \bar{\boldsymbol{\theta}}^t\|^2.
\tag{64}
$$

Plugging the update rule $\bar{\boldsymbol{\theta}}^{t+1} = \bar{\boldsymbol{\theta}}^t - \eta \boldsymbol{U}^t \mathbf{1}/K$, we have

$$
F(\bar{\boldsymbol{\theta}}^{t+1}) - F(\bar{\boldsymbol{\theta}}^t) \leq -\eta \langle \nabla F(\bar{\boldsymbol{\theta}}^t), \frac{\boldsymbol{U}^t \mathbf{1}}{K} \rangle + \frac{\eta^2 L_\mu}{2} \left\|\frac{\boldsymbol{U}^t \mathbf{1}}{K}\right\|^2.
\tag{65}
$$

Using the identity (36), we can write the first term of the left hand-side of (65) as

$$
\langle \nabla F(\bar{\boldsymbol{\theta}}^t), \frac{\boldsymbol{U}^t \mathbf{1}}{K} \rangle = \frac{1}{2}\left[\|\nabla F(\bar{\boldsymbol{\theta}}^t)\|^2 + \left\|\frac{\boldsymbol{U}^t \mathbf{1}}{K}\right\|^2 - \left\|F(\bar{\boldsymbol{\theta}}^t) - \frac{\boldsymbol{U}^t \mathbf{1}}{K}\right\|^2\right].
\tag{66}
$$

Next, we look for an upper bound for the third term of (66). To this end, we start by writing

$$\nabla F(\bar{\boldsymbol{\theta}}^t) - \frac{\boldsymbol{U}^t \mathbf{1}}{K}$$

$$= \frac{1}{\mu K} \sum_{i=1}^{K} \left\{ \exp\left(\frac{f_i(\bar{\boldsymbol{\theta}}^t)}{\mu}\right) \nabla f_i(\bar{\boldsymbol{\theta}}^t) - h(\boldsymbol{\theta}_i^t; \mu) g_i(\boldsymbol{\theta}_i^t) \right\}$$

$$= \frac{1}{\mu K} \sum_{i=1}^{K} \left\{ \exp\left(\frac{f_i(\bar{\boldsymbol{\theta}}^t)}{\mu}\right) \nabla f_i(\bar{\boldsymbol{\theta}}^t) - h(\bar{\boldsymbol{\theta}}^t; \mu) \nabla f_i(\bar{\boldsymbol{\theta}}^t) \right\} + \frac{1}{\mu K} \sum_{i=1}^{K} \left\{ h(\bar{\boldsymbol{\theta}}^t; \mu) \nabla f_i(\bar{\boldsymbol{\theta}}^t) - h(\boldsymbol{\theta}_i^t; \mu) \nabla f_i(\bar{\boldsymbol{\theta}}^t) \right\}$$

$$+ \frac{1}{\mu K} \sum_{i=1}^{K} \left\{ h(\boldsymbol{\theta}_i^t; \mu) \nabla f_i(\bar{\boldsymbol{\theta}}^t) - h(\boldsymbol{\theta}_i^t; \mu) g_i(\bar{\boldsymbol{\theta}}^t) \right\} + \frac{1}{\mu K} \sum_{i=1}^{K} \left\{ h(\boldsymbol{\theta}_i^t; \mu) g_i(\bar{\boldsymbol{\theta}}^t) - h(\boldsymbol{\theta}_i^t; \mu) g_i(\boldsymbol{\theta}_i^t) \right\}. \tag{67}$$

Using the inequality (35), we get

$$\left\| \nabla F(\bar{\boldsymbol{\theta}}^t) - \frac{\boldsymbol{U}^t \mathbf{1}}{K} \right\|^2$$

$$\leq \frac{4}{\mu^2 K} \sum_{i=1}^{K} \left\| \left( \exp\left(\frac{f_i(\bar{\boldsymbol{\theta}}^t)}{\mu}\right) - h(\bar{\boldsymbol{\theta}}^t; \mu) \right) \nabla f_i(\bar{\boldsymbol{\theta}}^t) \right\|^2 + \frac{4}{\mu^2 K} \sum_{i=1}^{K} \left\| \left( h(\bar{\boldsymbol{\theta}}^t; \mu) - h(\boldsymbol{\theta}_i^t; \mu) \right) \nabla f_i(\bar{\boldsymbol{\theta}}^t) \right\|^2$$

$$+ \frac{4}{\mu^2 K} \sum_{i=1}^{K} \left\| h(\boldsymbol{\theta}_i^t; \mu) \left( \nabla f_i(\bar{\boldsymbol{\theta}}^t) - g_i(\bar{\boldsymbol{\theta}}^t) \right) \right\|^2 + \frac{4}{\mu^2 K} \sum_{i=1}^{K} \left\| h(\boldsymbol{\theta}_i^t; \mu) \left( g_i(\bar{\boldsymbol{\theta}}^t) - g_i(\boldsymbol{\theta}_i^t) \right) \right\|^2. \tag{68}$$

Using assumption 2 and taking the expected value from both sides, we can write

$$\mathbb{E}\left[ \left\| \nabla F(\bar{\boldsymbol{\theta}}^t) - \frac{\boldsymbol{U}^t \mathbf{1}}{K} \right\|^2 \right]$$

$$\leq \frac{4G_1^2}{\mu^2 K} \sum_{i=1}^{K} \underbrace{\mathbb{E}\left[ \left| \exp\left(\frac{f_i(\bar{\boldsymbol{\theta}}^t)}{\mu}\right) - h(\bar{\boldsymbol{\theta}}^t; \mu) \right|^2 \right]}_{T_1} + \frac{4G_1^2}{\mu^2 K} \sum_{i=1}^{K} \underbrace{\mathbb{E}\left[ \left| h(\bar{\boldsymbol{\theta}}^t; \mu) - h(\boldsymbol{\theta}_i^t; \mu) \right|^2 \right]}_{T_2}$$

$$+ \frac{4(G_2(\mu))^2}{\mu^2 K} \sum_{i=1}^{K} \underbrace{\mathbb{E}\left[ \left\| \nabla f_i(\bar{\boldsymbol{\theta}}^t) - g_i(\bar{\boldsymbol{\theta}}^t) \right\|^2 \right]}_{T_3} + \frac{4(G_2(\mu))^2}{\mu^2 K} \sum_{i=1}^{K} \underbrace{\mathbb{E}\left[ \left\| g_i(\bar{\boldsymbol{\theta}}^t) - g_i(\boldsymbol{\theta}_i^t) \right\|^2 \right]}_{T_4}. \tag{69}$$

We start by bounding the term $T_1$

$$T_1 = \mathbb{E}\left[ \left| \exp\left(\frac{f_i(\bar{\boldsymbol{\theta}}^t)}{\mu}\right) - h(\bar{\boldsymbol{\theta}}^t; \mu) \right|^2 \right]$$

$$\overset{(15)}{\leq} L_\mu^2 \mathbb{E}\left[ \left| \frac{1}{B} \sum_{i=1}^{B} \frac{\ell(\bar{\boldsymbol{\theta}}^t, \xi_i^t)}{\mu} - \frac{f_i(\bar{\boldsymbol{\theta}}^t)}{\mu} \right|^2 \right]$$

$$= \frac{L_\mu^2}{B^2 \mu^2} \mathbb{E}\left[ \left| \sum_{i=1}^{B} \left( \ell(\bar{\boldsymbol{\theta}}^t, \xi_i^t) - f_i(\bar{\boldsymbol{\theta}}^t) \right) \right|^2 \right]$$

$$= \frac{L_\mu^2}{B^2 \mu^2} \mathbb{E}\left[ \left| \sum_{i=1}^{B} X_i^t \right|^2 \right]$$

$$= \frac{L_\mu^2}{B^2 \mu^2} \left( \sum_{i=1}^{B} \mathbb{E}\left[ |X_i^t|^2 \right] + \sum_{j \neq i} \mathbb{E}\left[ \langle X_i^t, X_j^t \rangle \right] \right), \tag{70}$$

where $X_i^t = \ell(\bar{\boldsymbol{\theta}}^t, \xi_i^t) - f_i(\bar{\boldsymbol{\theta}}^t)$. Since $\xi_i^t$ and $\xi_j^t$ are independent for $j \neq i$, then $\mathbb{E}\left[\langle X_i^t, X_j^t \rangle\right] = 0$. Then, using (25), we can write that

$$T_1 = \frac{L_\mu^2}{B^2 \mu^2} \sum_{i=1}^{B} \mathbb{E}\left[|X_i^t|^2\right] \leq \frac{L_\mu^2}{B^2 \mu^2}(B \sigma_\mu^2) = \frac{L_\mu^2 \sigma_\mu^2}{B \mu^2}. \tag{71}$$

Similarly, we can bound the term $T_3$ as $T_3 \leq \frac{\sigma_\mu^2}{B}$. Next, we bound the term $T_2$ as

$$\begin{aligned}
T_2 &= \mathbb{E}\left[\left|h(\bar{\boldsymbol{\theta}}^t; \mu) - h(\boldsymbol{\theta}_i^t; \mu)\right|^2\right] \\
&= \mathbb{E}\left[\left|\exp\left(\frac{1}{B}\sum_{j=1}^{B}\frac{\ell(\bar{\boldsymbol{\theta}}^t, \xi_j^t)}{\mu}\right) - \exp\left(\frac{1}{B}\sum_{j=1}^{B}\frac{\ell(\boldsymbol{\theta}_i^t, \xi_j^t)}{\mu}\right)\right|^2\right] \\
&\overset{(15)}{\leq} L^2 \mathbb{E}\left[\left|\frac{1}{B}\sum_{j=1}^{B}\frac{\ell(\bar{\boldsymbol{\theta}}^t, \xi_j^t)}{\mu} - \frac{1}{B}\sum_{j=1}^{B}\frac{\ell(\boldsymbol{\theta}_i^t, \xi_j^t)}{\mu}\right|^2\right] \\
&= \frac{L_\mu^2}{B^2 \mu^2}\mathbb{E}\left[\left|\sum_{j=1}^{B}(\ell(\bar{\boldsymbol{\theta}}^t, \xi_j^t) - \ell(\boldsymbol{\theta}_i^t, \xi_j^t))\right|^2\right] \\
&\leq \frac{L_\mu^2}{B \mu^2}\sum_{j=1}^{B}\mathbb{E}\left[\left|\ell(\bar{\boldsymbol{\theta}}^t, \xi_j^t) - \ell(\boldsymbol{\theta}_i^t, \xi_j^t)\right|^2\right] \\
&\overset{(a)}{\leq} \frac{L_\mu^2 G_\mu^2}{\mu^2}\mathbb{E}\left[\|\bar{\boldsymbol{\theta}}^t - \boldsymbol{\theta}_i^t\|^2\right],
\end{aligned} \tag{72}$$

where we have used the fact that (12) gives that $\ell(\bar{\boldsymbol{\theta}}^t, \xi_j^t)$ is $G$-Lipschitz continuous in (a) and (24) in the last inequality. Finally, using (11), we can bound $T_4$ as $T_4 \leq L_\mu^2 \mathbb{E}\left[\|\bar{\boldsymbol{\theta}}^t - \boldsymbol{\theta}_i^t\|^2\right]$. Using the bounds on the terms $T_1$-$T_4$, we can write

$$\mathbb{E}\left[\left\|\nabla F(\bar{\boldsymbol{\theta}}^t) - \frac{\boldsymbol{U}^t \mathbf{1}}{K}\right\|^2\right] \leq \frac{4G_\mu^2 \sigma_\mu^2(L_\mu^2 + \mu^2)}{\mu^4 B} + \frac{4G_\mu^2(G_\mu^2 + \mu^2)L_\mu^2}{\mu^4 K}\sum_{i=1}^{K}\mathbb{E}\left[\|\bar{\boldsymbol{\theta}}^t - \boldsymbol{\theta}_i^t\|^2\right]. \tag{73}$$

Replacing (73) in (66), we obtain

$$\mathbb{E}\left[\langle \nabla F(\bar{\boldsymbol{\theta}}^t), \frac{\boldsymbol{U}^t \mathbf{1}}{K}\rangle\right] \geq \frac{1}{2}\mathbb{E}\left[\|\nabla F(\bar{\boldsymbol{\theta}}^t)\|^2\right] + \frac{1}{2}\mathbb{E}\left[\left\|\frac{\boldsymbol{U}^t \mathbf{1}}{K}\right\|^2\right] - \frac{2G_\mu^2 \sigma_\mu^2(L_\mu^2 + \mu^2)}{\mu^4 B} - \frac{2G_\mu^2(G_\mu^2 + \mu^2)L_\mu^2}{\mu^4 K}\mathbb{E}\left[\|\boldsymbol{\theta}^t(\boldsymbol{I} - \boldsymbol{J})\|_F^2\right]. \tag{74}$$

Going back to (65), we can write

$$\mathbb{E}\left[F(\bar{\boldsymbol{\theta}}^{t+1}) - F(\bar{\boldsymbol{\theta}}^t)\right]$$
$$\leq -\frac{\eta}{2}\mathbb{E}\left[\|\nabla F(\bar{\boldsymbol{\theta}}^t)\|^2\right] + \frac{\eta}{2}(\eta L_\mu - 1)\mathbb{E}\left[\left\|\frac{\boldsymbol{U}^t \mathbf{1}}{K}\right\|^2\right] + \frac{2G_\mu^2 \sigma_\mu^2 \eta(L_\mu^2 + \mu^2)}{\mu^4 B} + \frac{4G_\mu^2 L_\mu^2 \eta(G_\mu^2 + \mu^2)}{\mu^4 K}\mathbb{E}\left[\|\boldsymbol{\theta}^t(\boldsymbol{I} - \boldsymbol{J})\|_F^2\right]. \tag{75}$$

Setting the learning rate $\eta$ such that $\eta L_\mu \leq 1$, and taking the average over $t \in [1, T]$, we get

$$\frac{\mathbb{E}\left[F(\bar{\boldsymbol{\theta}}^T) - F(\bar{\boldsymbol{\theta}}^1)\right]}{T} \leq -\frac{\eta}{2T}\sum_{t=1}^{T}\mathbb{E}\left[\|\nabla F(\bar{\boldsymbol{\theta}}^t)\|^2\right] + \frac{2\eta G_\mu^2 \sigma_\mu^2(L_\mu^2 + \mu^2)}{\mu^4 B} + \frac{2\eta G_\mu^2 L_\mu^2(G_\mu^2 + \mu^2)}{\mu^4 KT}\sum_{t=1}^{T}\mathbb{E}\left[\|\boldsymbol{\theta}^t(\boldsymbol{I} - \boldsymbol{J})\|_F^2\right]. \tag{76}$$

Re-arranging the terms and using assumption 5, we can write

$$\frac{1}{T}\sum_{t=1}^{T}\mathbb{E}\left[\|\nabla F(\bar{\boldsymbol{\theta}}^t)\|^2\right] \leq \frac{2(F(\bar{\boldsymbol{\theta}}^1) - F_{inf})}{\eta T} + \frac{2G_\mu^2\sigma_\mu^2(L_\mu^2 + \mu^2)}{\mu^4 B} + \frac{2G_\mu^2 L_\mu^2(G_\mu^2 + \mu^2)}{\mu^4 KT}\sum_{t=1}^{T}\mathbb{E}\left[\|\boldsymbol{\theta}^t(\boldsymbol{I} - \boldsymbol{J})\|_F^2\right].$$

(77)

Using Lemma 3 in (77), we get

$$\frac{1}{T}\sum_{t=1}^{T}\mathbb{E}\left[\|\nabla F(\bar{\boldsymbol{\theta}}^t)\|^2\right] \leq \frac{2(F(\bar{\boldsymbol{\theta}}^1) - F_{inf})}{\eta T} + \frac{2G_\mu^2\sigma_\mu^2(L_\mu^2 + \mu^2)}{\mu^4 B} + \frac{4\eta^2 L_\mu^2\rho G_\mu^2(G_\mu^2 + \mu^2)[8\sigma_\mu^2(L_\mu^2 + 1) + G_\mu^2]}{\mu^8(1 - \gamma)(1 - \sqrt{\rho})^2}.$$

(78)

Furthermore, choosing $\eta$ such that $\eta L_\mu < \frac{\mu(1 - \sqrt{\rho})}{8G_\mu\sqrt{\rho}}$ ensures that $\gamma_\mu < \frac{1}{4}$, then we can write

$$\frac{1}{T}\sum_{t=1}^{T}\mathbb{E}\left[\|\nabla F(\bar{\boldsymbol{\theta}}^t)\|^2\right] \leq \frac{2(F(\bar{\boldsymbol{\theta}}^1) - F_{inf})}{\eta T} + \frac{2G_\mu^2\sigma_\mu^2(L_\mu^2 + \mu^2)}{\mu^4 B} + \frac{16\eta^2 L_\mu^2\rho G_\mu^2(G_\mu^2 + \mu^2)[8\sigma_\mu^2(L_\mu^2 + 1) + G^2]}{3\mu^8(1 - \sqrt{\rho})^2},$$

(79)

which finalizes the proof.

