# OpenReview forum: "DR-DSGD: A Distributionally Robust Decentralized Learning Algorithm over Graphs"
_TMLR — Accepted by TMLR_

### Review · Reviewer_A9Fv · 2022-05-10

**Summary Of Contributions:**

The paper proposed a decentralized optimization algorithm for agnostic federated learning. The proposed algorithm is essentially decentralized SGD applied to a slightly different formulation than empirical risk minimization. Experiments on Fashion MNIST and CIFAR-10 are done to compare the proposed algorithm with decentralized SGD. Empirically, it is shown the proposed algorithm can converge faster and can achieve smaller accuracy variance across nodes.

**Broader Impact Concerns:**

No concerns.

**Requested Changes:**

1. If the analyses is just redoing traditional analyses for decentralized SGD, please remove it and cite existing results. If not, please highlight the difference and this could be part of the contribution.

2. Please include parameter tuning procedures like how stepsizes and \mu are chosen for the experiments.


**Strengths And Weaknesses:**

Strengths:
1. The proposed algorithm has better performance than decentralized SGD
2. The proposed algorithm has rigorous convergence guarantees since it is essentially decentralized SGD.

Weakness:
1. The analysis seems to be unnecessary since there are many existing analyses for decentralized SGD.
2. The experiment is a bit weak and unconvincing for several reasons. 1). The parameter tuning procedure seems to be missing in the paper. 2). The experiments use small datasets, a small neural network, and a small number of devices. It is hard to say how the algorithm will perform in more practical settings.

---

> ### Author Response · Authors · 2022-05-30
> **Reply to Reviewer A9Fv**
>
> **Comment:**
> > If the analyses is just redoing traditional analyses for decentralized SGD, please remove it and cite existing results. If not, please highlight the difference and this could be part of the contribution.
>
> **Reply:**
> We would like to clarify that the objective of the paper is to propose a framework for **robust** decentralized learning. Therefore; as highlighted in Remark 1, our convergence analysis involves **biased** stochastic gradients due to the compositional nature of the reformulated loss function given in (7). This is in contrast to existing FL frameworks that rely on the **unbiasedness** of the stochastic gradients. This makes the analysis more challenging and different from the traditional analyses for decentralized SGD. As per the reviewer’s suggestion, we include this statement in the contribution part of the revised manuscript.
> ---
> **Comment:**
> > The experiments use small datasets, a small neural network, and a small number of devices. It is hard to say how the algorithm will perform in more practical settings.
>
> **Reply:**
> In the experiments section, we used Fashion MNIST and CIFAR10 datasets, which we believe are datasets of moderate size (Fashion MNIST: a training set of 60000 examples and a test set of 10000 examples and CIFAR10: a training set of 60000 examples and a test set of 10000 examples) and often widely used, kindly refer to Mohri et al., 2019; Reisizadeh et al., 2020; and Deng et al., 2020, for instance. For the neural network (NN) choice, we believe that the comparison is fair as long as we are using the same NN architecture for both algorithms. The number of parameters of the NN used for Fashion MNIST is 109386, while the one used for CIFAR10 is 541094. We think both architectures are of moderate size and sufficient for illustration purposes. Furthermore, we believe that using a bigger NN
> architecture will improve the accuracy of both algorithms but will not alter the conclusions drawn in the paper. For the number of devices, we consider K = 10 in Section 6.2 and K = 25 in the rest of the sections of the experiments. We would like to note that the choice of a relatively small number of devices increases data heterogeneity. In fact, works dealing with robustness in FL (Reisizadeh et al., 2020; and Deng et al., 2020) use a similar number of devices.
> ---
> **Comment:**
> > Please include parameter tuning procedures like how stepsizes and $\mu$ are chosen for the experiments.
>
> **Reply:**
> Unless explicitly stated (when we study the impact of $\mu$ on the fairness and average test accuracy in Section 6.4.), we choose the learning rate $\eta$ and the regularization parameter $\mu$ according to Corollary 1. To make it clearer, we added the above statement in Section 6.1.

---

> > ### Comment · Reviewer_A9Fv · 2022-06-11
> > **Could you clarify more on how hyperparameters are chosen?**
> >
> > I see now that the bias caused by the minibatch sampling of training data within clients is the main difference compared with traditional unbiased analyses. And the bias is controlled by minibatch size. Thanks for pointing this out.
> >
> > As for the hyperparameter tuning procedure, could you clarify more about it? The current version of Corollary 1 is on how to choose \eta to achieve the best convergence rate given T, L, and K. Then how T is chosen and how L is estimated? Also, Corollary 1 seems to miss mentioning how \mu is chosen?

---

> > > ### Author Response · Authors · 2022-06-13
> > > **Reply**
> > >
> > > We thank the reviewer for the comment. Based on our discussion with reviewer Xb1C, we realized that $\mu$ is part of the objective function. Hence, $\mu$ cannot be chosen to our liking, and the convergence rate needs to be derived for any value of $\mu$. This is primarly why we modified part of the proof, including Corollary 1, where now we only choose the learning rate $\eta$ and mini-batch size $B$ since both are parts of the algorithm. The number of iterations $T$ is a parameter that needs to be fixed before running the algorithm. Finally, $L$ can be estimated in a similar way as in ([Fazlyab et al, 2019](https://proceedings.neurips.cc/paper/2019/file/95e1533eb1b20a97777749fb94fdb944-Paper.pdf), [Latorre et al., 2020](https://openreview.net/pdf?id=rJe4_xSFDB)). Note that in the original version, we don't require the estimation of $L$; however, to ensure that the condition of the learning rate introduced in Theorem 1 is satisfied, we introduced the current express of $\eta$. As $T \rightarrow \infty$, both choices become almsost the same.

---

> > > > ### Comment · Reviewer_A9Fv · 2022-06-13
> > > > **How about the parameter tuning process in the experiments?**
> > > >
> > > > Thank you for explaining the changes, it indeed makes more sense to me after the explanation on \mu. However, I am and have been actually wondering how the authors have set the \eta and \mu in the experiments in the paper, but not the theoretically optimal values. Could you elaborate more on how the parameters are chosen in experiments? If the experiments used theoretically suggested values, will changing corollary 1 undermine the validity of the experiments?

---

> > > > > ### Author Response · Authors · 2022-06-17
> > > > > **Reply**
> > > > >
> > > > > We thank the reviewer for the follow-up question. We added a sentence explaining the choice of $\eta$ and $B$ in Section 6.1. In fact, we are using the learning rate $\eta = \sqrt{K/T}$ without the additional estimation of $L$ (as this happens to be our original setting). We also specify the value of $\mu$ in every subsection of the experiments section. Currently, we simply choose a value that is not very small (in order not to violate some of the assumptions where making), and also not a high value, as it approaches the original ERM minimization problem (in other words, less robustness). Furthermore, we investigate the effect of $\mu$ in Section 6.4

---

### Review · Reviewer_7oCx · 2022-05-13

**Summary Of Contributions:**

This paper proposes and analyzes a new method, distributionally-robust decentralized stochastic gradient descent or DR-DSGD, for robust decentralized learning. The target setting is motivated by federated learning, where there is data heterogeneity. Rather than the usual objective, which corresponds to minimizing the empirical risk, the proposed objective is the usual one considered in distributionally robust optimization, where the aim is to minimize the loss wrt the the worst-case mixture over data distributions. Using existing ideas from the literature, this min-max problem is transformed to a minimization problem with a different objective, and the standard DSGD method is applied to the new formulation. A convergence analysis is provided for the smooth non-convex setting, and experiments validate the convergence analysis, highlighting the promising performance of the proposed approach in comparison to standard DSGD applied to the ERM formulation.

**Broader Impact Concerns:**

I don't have any major concerns about ethical implications of the work. At the same time, given that the paper addresses fairness and robustness, it seems like there is a missed opportunity here to discuss the potential broader impact of this work, since the Broader Impact Statement has been omitted in the initial submission.

**Requested Changes:**

The suggested changes here are primarily to address weaknesses mentioned above.

### Accuracy of several statements in the intro
There are several statements made in the introduction which are highly debatable and should be better supported by evidence.
* "privacy ensured by only exchanging models/gradients with server" This is not true; see for example, work on gradient inversion attacks such as [Geiping et al. 2003](https://arxiv.org/abs/2003.14053). Additional mechanisms are needed (differential privacy, secure aggregation) to formally ensure privacy.
* "Most FL algorithms fail to address the data heterogeneity issue" There have been several methods proposed in the FL literature that aim to address the data heterogeneity issue, and which ought to be better acknowledged. For example, see FedProx, VRLSGD, Scaffold, Mime, FedNova, FedLin, and FedShuffle.
* Although I agree that FedAvg-style methods are state-of-the-art, I would not call FedAvg itself state-of-the-art in FL. Again, see the methods mentioned in the previous point, which exhibit superior performance to FedAvg.

### Relationship to other literature on robustness
Several other papers in the literature address robustness in FL in different senses. While the paper does already mention several relevant references, there are a few others I was also expecting to see:
* [Pillutla et al., 2019](https://arxiv.org/abs/1912.13445) and references therein
* [Li et al., 2020](https://arxiv.org/abs/2012.04221)

### Relationship to other decentralized algorithms for distributionally-robust optimization
There is some prior literature looking at decentralized distributionally-robust optimization specifically in the context of power systems. It would be good to call out this connection and also acknowledge prior work. For example, see
* https://ieeexplore.ieee.org/abstract/document/9542394
* https://ieeexplore.ieee.org/document/9208508

### Explaining, justifying, and possibly simplifying assumptions
Presently, the analysis in Section 5 requires some strong assumptions, and it is not clear whether these are all necessary.
* Assumption 1, equation (12) essentially follows from assuming that the range of the ($\mu$-scaled) loss function is bounded. Can you provide examples of loss functions that satisfy this assumption? The only way I can see to ensure this for most commonly-used losses is to impose a clipping. If that's indeed necessary, then it would be better to state it explicitly. Once you assume that the range is bounded, then smoothness of $F_i$ and $F$ follows directly from smoothness of $f_i$. Often one may already know or more easily check properties of $\ell$, and hence $f_i$, than properties of $F_i$ or $F$.
* Equation (11) is not well-defined as written, since $g_i$ are stochastic gradients. Is there a missing expectation? Or is the assumption really that $\| \nabla \ell(\theta_1, \xi_i^t) - \nabla \ell(\theta_2, \xi_i^t)\| \le L_1 \|\theta_1 - \theta_2\|$ for all $\xi_i^t$?
* The bounded gradient assumption is rather strong. Can you provide examples where this holds?

### More general suggestions about the presentation of the theoretical results
* The ultimate objective (8) on which the DR-DSGD method is based is the composition of the original loss $f_i$ with the exponential function, which is smooth, convex, and non-decreasing. I appreciate that the analysis focuses on guarantees for non-convex functions. However, to start, it could be worthwhile to point out that if $f_i$ is convex (e.g., if $\ell$ is convex), then convergence of the proposed method follows directly from existing results in the literature. In particular, since $\exp(\cdot)$ is convex and non-decreasing, if $f_i$ is also convex then (8) is convex too.
* In the non-convex case, as mentioned in Remark 1, gradients aren't unbiased. It could be worth pointing out that this is not necessarily detrimental, and it is well known (e.g., [Bottou et al. (2016)](https://arxiv.org/abs/1606.04838) Theorems 4.8 and 4.9) that stochastic gradient methods can still converge as long as the stochastic gradients are sufficiently well-aligned with the expected gradient.
* How are $\sigma$, $L$, and $G$ in Lemma 2 defined? The definitions ought to be included in the main body of the paper for readability.

### Suggestions for the experiments section
* Please include more details about the ``pathological non-IID way'' in which data is distributed across devices, to enable an interested reader to reproduce your results.
* Was it expected that DR-DSGD improves Average Test Accuracy, in addition to Worst Distribution Test Accuracy? Or is this a pleasant consequence? Is there some intuition for why this is the case? Would it be the same if data were IID across devices?
* The empirical comparison only is made between DR-DSGD and DSGD. It would significantly strength the work and increase the interest of the paper if a comparison was included to other methods from the literature on robustness and fairness in FL, such as DITTO, Tilted empirical risk minimization, and DRFA.
* The `networkx` package provides functions to generate several types of random graphs. It sounds like the ones used in the experiment are Erdos-Renyi. Is that correct? Please clarify. It could be relevant to explore performance on several other types of graphs (e.g., rings, grids, ...) to validate the theoretical dependence on network topology.

**Strengths And Weaknesses:**

## Strengths
* The paper is generally well-written and pleasant to read
* Experimental results are promising, showing consistent improvements over plain DSGD
* The theoretical analysis guaranteeing convergence at the expected $\mathcal{O}(1/KT)$ rate appears to be correct.

## Weaknesses
I elaborate on each of the following weaknesses below, when discussing requested changes to address them:
* Several statements in the introduction are highly debatable. The paper would be strengthened by providing better support for these, or by softening the statements.
* There are some relevant references on robustness in FL that are missing
* There are relevant references to related work on decentralized distributionally-robust methods that are missing
* Several assumptions in the analysis should be more clearly explained, better justified, or possibly even simplified
* There are ways in which the presentation of theoretical results in Section 5 could be improved
* The experimental evaluation only compares DR-DSGD with DSGD. In particular, there isn't any comparison to other methods in the literature that aim to address robustness or data heterogeneity.

---

> ### Author Response · Authors · 2022-06-01
> **Reply to Reviewer 7oCx (Part 1)**
>
> ### 1. Accuracy of several statements in the intro:
> **Comment:**
> > "privacy ensured by only exchanging models/gradients with server". This is not true; see for example, work on gradient inversion attacks such as Geiping et al. 2003. Additional mechanisms are needed (differential privacy, secure aggregation) to formally ensure privacy.
>
> **Reply:** We agree with the reviewer. Though FL hides the raw data, it has been shown that gradient inversion attacks can still infer some information about the input. Hence, formally speaking, privacy is not ensured by just using FL. However, FL with an additional privacy-preservation mechanism can ensure privacy at a much lower communication cost than applying such a mechanism to the raw data. To address this, we rewrite the statement as: ''FL hides the raw data, so it can significantly reduce the communication cost. When combined with some privacy-preservation mechanism, it further ensures privacy.''
>
> ---
> **Comment:**
> > "Most FL algorithms fail to address the data heterogeneity issue" There have been several methods proposed in the FL literature that aim to address the data heterogeneity issue, and which ought to be better acknowledged. For example, see FedProx, VRLSGD, Scaffold, Mime, FedNova, FedLin, and FedShuffle.
>
> **Reply:** We agree with the reviewer. FedAvg was shown to fail to address the data heterogeneity issue. Various algorithms, including the ones mentioned by the reviewer, have been proposed to solve this issue for the empirical risk minimization problem. Instead, in our work, we propose a framework that solves this issue in the decentralized robust learning setting. To address this comment, we added the following text to the revised manuscript: ''Training using FedAvg presents several challenges that need to be tackled. It often fails to address the data heterogeneity issue. Though many enhanced variants of FedAvg (Li et al., 2020b; Liang
> et al., 2019; Karimireddy et al., 2020b;a; Wang et al., 2020; Mitra et al., 2021; Horváth et al., 2022) were proposed to solve this issue in the mean risk minimization problem, local data distributions might differ greatly from the average distribution. Therefore, a considerable drop in the global model performance on local data is often seen, suggesting that the mean risk minimization may not be the right objective.''
>
> ---
> **Comment:**
> > Although I agree that FedAvg-style methods are state-of-the-art, I would not call FedAvg itself state-of-the-art in FL. Again, see the methods mentioned in the previous point, which exhibit superior performance to FedAvg.
>
> **Reply:** We agree with the reviewer. To address this comment, we rewrite the statement as "FedAvg-style (FedAvg)-style methods (McMahan et al.,2017; Li et al., 2020b; Wang et al., 2020) are state-of-the-art".
>
> ---
> ## 2. Relationship to other literature on robustness:
> **Comment:**
> > Several other papers in the literature address robustness in FL in different senses. While the paper does already mention several relevant references, there are a few others I was also expecting to see: Pillutla et al., 2019 and references therein and Li et al., 2020.
>
> **Reply:** Thanks for pointing out some interesting related works. We have included them in the revised version. In particular, we added the following text: ''The work in (Pillutla et al, 2020) was proposed to tackle the problem of robustness to corrupted updates by applying robust aggregation based on the geometric median. Moreover, robustness to data and model poisoning attacks was also investigated by (Li et al, 2020)."
>
> ---
> ## 3. Relationship to other decentralized algorithms for distributionally robust optimization:
> **Comment:**
> > There is some prior literature looking at decentralized distributionally robust optimization specifically in the context of power systems. It would be good to call out this connection and also acknowledge prior work. For example, see: https://ieeexplore.ieee.org/abstract/document/9542394 and https://ieeexplore.ieee.org/document/9208508.
>
> **Reply:** Thanks for pointing out some interesting related works, which we have included in the revised version. In particular, we added the following text: ''DRO has also been applied in multi-regional power systems to improve the reliability by considering the uncertainty of wind power distributions and constructing a multi-objective function that maintains a trade-off between the operational cost and risk (Li \& Yang, 2020; Hu et al., 2021)."

---

> > ### Comment · Reviewer_7oCx · 2022-06-06
> > **Reply to Part 1 - Please clarify difference from previous work on decentralized DRO**
> >
> > Thank you for these responses. They mostly address the concerns mentioned (1--3). Regarding the last point, I appreciate the addition of this sentence. In addition to clarifying the different setting (power systems vs machine learning), it would be good to clarify what are the key differences (if any) between the methods proposed in those papers and the DR-DSGD method.

---

> > > ### Author Response · Authors · 2022-06-07
> > > **Reply**
> > >
> > > We thank the reviewer for the suggestion. We added the following sentence: "However, none of these works provides any convergence guarantees. Furthermore, our analysis is different from these works since it involves biased stochastic gradients."

---

> ### Author Response · Authors · 2022-06-01
> **Reply to Reviewer 7oCx (Part 2)**
>
> ## 4. Explaining, justifying, and possibly simplifying assumptions:
> **Comment:**
> >Assumption 1, equation (12) essentially follows from assuming that the range of the (-scaled) loss function is bounded. Can you provide examples of loss functions that satisfy this assumption? The only way I can see to ensure this for most commonly-used losses is to impose a clipping. If that's indeed necessary, then it would be better to state it explicitly. Once you assume that the range is bounded, then smoothness of $F_i$ and $F$ follows directly from smoothness of $f_i$. Often one may already know or more easily check properties of $\ell$, and hence $f_i$, than properties of $F_i$ or $F$.
>
> **Reply:** First of all, we agree with the reviewer's comment. For that, we added the following statement in the revised manuscript: ``Note that (12) can be fulfilled by imposing loss clipping (Xu et al., 2006; Wu \& Liu, 2007; Yang et al., 2010) to most commonly-used loss functions.'' Second, we would like to mention that in our current simulation setting, we haven't used loss clipping. It is true that the categorical cross-entropy function is indeed not bounded upwards. However, it will only take on large values if the predictions are very wrong. Let's first examine the behavior of a randomly initialized network. With random weights, the many units/layers will usually compound to result in the network outputting approximately uniform predictions. In a classification problem with $M$ classes, we will get probabilities of around $1/M$ for each category. Under some mild assumptions, the categorical cross-entropy will be around the entropy of a $M$-class uniform distribution, which is $\log(M)$. In fact, the cross-entropy for a single data point is $-\sum_{m=1}^M y_m \log(\hat{y}_m)$ where $y$ are the true probabilities (labels), $\hat{y}$ are the predictions. With hard labels (i.e. one-hot encoded), only a single $y_m$ is 1, and all others are 0. Thus, the term reduces to $-\log(\hat{y}_m)$, where $m$ is now the correct class. With a randomly initialized network, we have $\hat{y}_m \sim 1/M$, therefore, we get $-\log(1/M) = \log(M)$. Since the training objective is usually to reduce cross-entropy, we can think of $\log(M)$ as a worst-case value. If it ever gets higher, there is probably something wrong with the model. Considering a relatively moderate number of classes, e.g., $M=100$, we get $\log(M) < 5$. Thus, the (-scaled) cross-entropy will generally be relatively small in practice, and the assumption will hold in practice.
>
> ---
> **Comment:**
> >Equation (11) is not well-defined as written, since $g_i$ are stochastic gradients. Is there a missing expectation? Or is the assumption really that $| \nabla \ell(\theta_1, \xi_i^t) - \nabla \ell(\theta_2, \xi_i^t)| \le L_1 |\theta_1 - \theta_2|$ for all $\xi_i^t$?
>
> **Reply:** We thank the reviewer for pointing this out. Indeed, there is a missing expectation. We fixed this in the revised manuscript.
>
> ---
> **Comment:**
> >The bounded gradient assumption is rather strong. Can you provide examples where this holds?
>
> **Reply:** We agree with the reviewer that the bounded gradient assumption is rather strong. However, we would like to mention that such an assumption has been widely used in previous works focusing on robustness in FL, e.g. Mohri et al., 2019; Li et al., 2020d; and Deng et al., 2020, for instance, and is needed for the current convergence analysis. The relaxation of the uniform boundedness of the gradient is left for future work (we have added Section 6.6 Limitations where we list some of the limitations of the current work).
>
> ---
> ## 5. More general suggestions about the presentation of the theoretical results:
> **Comment:**
> >The ultimate objective (8) on which the DR-DSGD method is based is the composition of the original loss $f_i$ with the exponential function, which is smooth, convex, and non-decreasing. I appreciate that the analysis focuses on guarantees for non-convex functions. However, to start, it could be worthwhile to point out that if $f_i$  is convex (e.g., if $\ell$ is convex), then convergence of the proposed method follows directly from existing results in the literature. In particular, since $\exp(\cdot)$ is convex and non-decreasing, if $f_i$  is also convex then (8) is convex too.
>
> **Reply:** We agree with the reviewer. As pointed out, the focus of the paper is on the non-convex case. Taking into account the reviewer's comment, we added the following remark at the beginning of the convergence analysis section ``It is worth mentioning that since the exponential function is convex, and non-decreasing function, then $\exp(f_i(\cdot))$ is convex when $f_i(\cdot)$ is convex. In this case, the convergence of the proposed method follows directly from existing results in the literature (Yuan et al, 2016).''.

---

> > ### Comment · Reviewer_7oCx · 2022-06-06
> > **Part 2 - Thank you**
> >
> > Thank you, your replies address these comments (4--5) well. I encourage you to include the discussion around $\log(M)$ as a typical worst-case bound in the paper to help justify this to readers.

---

> > > ### Author Response · Authors · 2022-06-07
> > > **Reply**
> > >
> > > We thank the reviewer for the suggestion. We have added a brief discussion on the worst-case bound for the case of categorical cross-entropy in the main body, and we referred the reader to Appendix A.1. for the full explanation.

---

> ### Author Response · Authors · 2022-06-01
> **Reply to Reviewer 7oCx (Part 3)**
>
> ## 5. More general suggestions about the presentation of the theoretical results:
> **Comment:**
> >In the non-convex case, as mentioned in Remark 1, gradients aren't unbiased. It could be worth pointing out that this is not necessarily detrimental, and it is well known (e.g., Bottou et al. (2016) Theorems 4.8 and 4.9) that stochastic gradient methods can still converge as long as the stochastic gradients are sufficiently well-aligned with the expected gradient.
>
> **Reply:** We thank the reviewer for pointing us toward the reference (Bottou et al. (2016) Theorems 4.8 and 4.9). If our understanding is correct, this is related to Assumption 4.3 (b). First, we do not require such a condition to hold in our case, so in that sense, our analysis is more general. Furthermore, the analysis in (Bottou et al. 2016) covers the choices of stochastic gradient mentioned in (Bottou et al. (2016), Eq. (4.2)). The first two represent  \textbf{unbiased} estimators, while the third choice requires $H_k$ to be a positive definite matrix that is \textbf{conditionally uncorrelated with} $g(w_k, \xi_k)$ given $w_k$ and whose eigenvalues lie in a fixed positive interval for all $k \in \mathbb{N}$. Clearly, these choices do not cover our setting.
>
> ---
> **Comment:**
> >How are $\sigma$, $L$, and $G$ in Lemma 2 defined? The definitions ought to be included in the main body of the paper for readability.
>
> **Reply:** We thank the reviewer for pointing this out. The definitions of $\sigma$, $L$, and $G$ are now defined in the main body of the paper, right after Lemma 2.
>
> ---
> ## 6. Suggestions for the experiments section:
> **Comment:**
> >Please include more details about the ''pathological non-IID way'' in which data is distributed across devices, to enable an interested reader to reproduce your results.
>
> **Reply:** More details about  the ''pathological non-IID way'' in which data are distributed across devices are now included in Section 6.1. The following statement is added to the revised manuscript: ''More specifically, we first order the samples according to the labels, and then we divide the data into shards of equal sizes. Finally, we assign each device the same number of chunks. This will ensure a pathological non-IID partitioning of the data, as most devices will only have access to certain classes and not all of them.''
>
> ---
> **Comment:**
> >Was it expected that DR-DSGD improves Average Test Accuracy, in addition to Worst Distribution Test Accuracy? Or is this a pleasant consequence? Is there some intuition for why this is the case? Would it be the same if data were IID across devices?
>
> **Reply:** We thank the reviewer for pointing out this point. Upon investigating this behaviour, we found a ''bug'' in the computation of the average test accuracy for DSGD. We fixed this issue in the revised manuscript. We can see that now DR-DSGD has almost a similar performance as DSGD in terms of the average test accuracy.
>
> ---
> **Comment:**
> >The empirical comparison only is made between DR-DSGD and DSGD. It would significantly strength the work and increase the interest of the paper if a comparison was included to other methods from the literature on robustness and fairness in FL, such as DITTO, Tilted empirical risk minimization, and DRFA.
>
> **Reply:** We thank the reviewer for the suggestion. We think that since our work focuses on the decentralized setting, making a comparison with other FL works (e.g., DITTO, Tilted empirical risk minimization, and DRFA) that rely on a parameter-server topology would not fair.
>
> ---
> **Comment:**
> >The networkx package provides functions to generate several types of random graphs. It sounds like the ones used in the experiment are Erdos-Renyi. Is that correct? Please clarify. It could be relevant to explore performance on several other types of graphs (e.g., rings, grids, ...) to validate the theoretical dependence on network topology.
>
> **Reply:** First, we confirm that the networks used in our experiments are indeed Erdős-Rényi graphs or binomial graphs. To make it clear, we added the following statement in Section 6.1: ``Unless otherwise stated, the graphs used in our experiments are of Erdős-Rényi type.'' Second, we thank the reviewer for the suggestion. In the revised manuscript (Section 6.5), we explore the performance of both algorithms on several other types of graphs (specifically rings, grids, and geometric ones) in Figure 6.
>
> ---
> ## 7. Broader Impact Concerns:
> **Comment:**
> >I don't have any major concerns about the ethical implications of the work. At the same time, given that the paper addresses fairness and robustness, it seems like there is a missed opportunity here to discuss the potential broader impact of this work, since the Broader Impact Statement has been omitted in the initial submission.
>
> **Reply:** We thank the reviewer for pointing this out. In the revised manuscript, we have added Section 8. Broader Impact Statement.

---

> > ### Comment · Reviewer_7oCx · 2022-06-06
> > **Responses - Part 3**
> >
> > Thank you for these responses and for adding the broader impact statement.
> >
> > Regarding the bug in accuracy computation, can you elaborate on why it only affected the results of DSGD and not DR-DSGD? Some additional discussion would be useful to restore confidence in the other experimental results. Have you committed to releasing code to reproduce the experimental results?
> >
> > Part of the reason I ask is that it is somewhat surprising that DR-DSGD is able to match the results of DSGD in terms of Average Test Accuracy while also achieving much better Worst-Distribution Test Accuracy. I would have expected that the improvement in Worst-Distribution Test Accuracy would be accompanied by some decrease in Average Test Accuracy. It would be great to understand/justify this phenomenon better.
> >
> > Regarding the comparison to other methods, although the communication pattern of most FL methods involves communication with a server, the action of the server is to average updates from the clients and return a new model. When all clients participate in every round, this is mathematically equivalent to running a decentralized method (with local updates) over a complete graph. Hence, couldn't a fair comparison could be made at least in the specific case of a complete graph topology?
> >
> > I acknowledge that the main contributions of the paper are the theoretical convergence analysis, but I believe the experiments serve an important point of illustrating the promise of the proposed approach and motivating others to adopt or otherwise follow-up on this work.

---

> > > ### Author Response · Authors · 2022-06-07
> > > **Reply**
> > >
> > > **Comment:**
> > > > Regarding the bug in accuracy computation, can you elaborate on why it only affected the results of DSGD and not DR-DSGD? Some additional discussion would be useful to restore confidence in the other experimental results. Have you committed to releasing code to reproduce the experimental results? Part of the reason I ask is that it is somewhat surprising that DR-DSGD is able to match the results of DSGD in terms of Average Test Accuracy while also achieving much better Worst-Distribution Test Accuracy. I would have expected that the improvement in Worst-Distribution Test Accuracy would be accompanied by some decrease in Average Test Accuracy. It would be great to understand/justify this phenomenon better.
> > >
> > > **Reply:** We thank the reviewer for the comment. We are implementing the DSGD and DR-DSGD in two different Jupyter notebooks. That is why the bug we found affects only the implementation of DSGD. We plan to put our code on Github, as this makes reproducing the paper's results easier for the reader. As for the performance of DR-DSGD compared to DSGD, in terms of the average test accuracy, we would like to point out that similar behaviour is seen in (Deng et al., 2020, Fig. 3), where DRFA, AFL, and q-Fedavg have a similar performance to Fedavg. Similarly, in (Li et al, 2020d), q-FFL (q > 0) have a similar average test performance to q-FFL (q = 0), kindly refer to Tables 6 and 7. Finally, TERM introduced in (Li et al., 2021a) also achieves the same average test accuracy as FedAvg (see Fig. 16 and Table 7). We added the following sentence to explain this behaviour: "This is mainly due to the fact that while the ERM objective
> > > sacrifices the worst-case performance for the average one, DRO aims to lower the variance while maintaining a good average performance across the different devices."
> > >
> > > ---
> > > **Comment:**
> > > > Regarding the comparison to other methods, although the communication pattern of most FL methods involves communication with a server, the action of the server is to average updates from the clients and return a new model. When all clients participate in every round, this is mathematically equivalent to running a decentralized method (with local updates) over a complete graph. Hence, couldn't a fair comparison could be made at least in the specific case of a complete graph topology? I acknowledge that the main contributions of the paper are the theoretical convergence analysis, but I believe the experiments serve as an important point of illustrating the promise of the proposed approach and motivating others to adopt or otherwise follow up on this work.
> > >
> > > **Reply:** We thank the reviewer for the comment. We will work on adding a section where we compare some of the baselines mentioned by the reviewer to DR-DSGD when considering the case of a complete graph while we restrict the local updates to one iteration to make the comparison fairer.

---

### Review · Reviewer_Xb1C · 2022-05-24

**Summary Of Contributions:**

This works studies distributionally robust optimization, i.e., the problem of minimizing the worst linear combination of given losses $f_i$, where $f_i$ is the objective of $i$-th device. The authors assume that $f_i$ is given in the form of expectation over random samples, with bounded values, bounded variance of values, and bounded variance of the gradients as well.
The authors formulate this problem and then relax it using Kullback-Leibler regularization from uniform weights. They parameterize the regularization with coefficient $\mu$, which gives the original problem if $\mu=0$ and gives standard empirical risk if $\mu\to+\infty$. The last objective can be equivalently rewritten as $\min_{\Theta} \frac{1}{K}\sum_{i=1}^k F_i(\Theta)$, where $F_i(\Theta)=\frac{\exp(f_i(\Theta))}{\mu}$.
The main contribution is a decentralized algorithm that runs stochastic gradient descent on each device and communicates with neighbors given by a mixing matrix $W$. The authors state its convergence in Theorem 1, which is roughly saying that after $T$ iterations we have $\mathbb{E}[\Vert\nabla F(\overline \theta)\Vert^2] = O\left(\frac{1}{T} + \frac{1}{\mu^2} \right)$.
The authors also run experiments to compare their algorithm DR-DSGD with non-decentralized DSGD.

**Broader Impact Concerns:**

I do not think that there might be any ethical concern about this work.

**Requested Changes:**

The main change required from the authors is to fix convergence and dependence on $\mu$ in the constants. There should also be a discussion of the assumptions as they are quite restrictive.

## Minor points
1. One thing that really confuses me is that superscripts are used both as iteration counters and matrix powers, sometimes within the same equation, for instance in (36).
2. The authors claim several times that their algorithm is "robust to data heterogeneity". I am not sure what this means and where it is shown. Instead, it seems that the algorithm is robust to distribution shift, which is not the same as robust to higher data heterogeneity. Moreover, I suspect that any algorithm that tries to use distributional robustness would be less robust to data heterogeneity, as it may largely overfit to the data on a single client.
3. Why is DR-DSGD better than DSGD in the experiments in terms of average test accuracy? From what I understand, DR-DSGD is solving a harder problem, so it is natural to expect that it would be better in terms of worst-distribution test accuracy, but the improvement of the average accuracy is quite surprising. For instance, in the work on "tilte" risk minimization, Table 7, only an improvement for the worst devices is reported.

## Typos
Page 2, "on distributionally robustness" -> "on distributional robustness"
Page 2, "as shown in corollary 1" -> "as shown in Corollary 1"
Page 3, "set of neighbors of device $n$ is defined as $N_i$" -> "set of neighbors of device $i$ is defined as $N_i$"
Page 3, "if only if" -> "if and only if"
Page 4, the text writes "a local stochastic gradient update (Line 3) using the learning rate $\eta_t$", but in the algorithm itself $\eta$ is used without a subscript
Page 5, equation (15), it's not vectors, so norms need to be replaced with absolute values
Assumption 4 should probably mention that $F_{inf}>-\infty$
Page 7, "we start by stating a key lemma". It's a bit strange to write this after stating Lemma 1
Page 7, "Let $\eta$ satisfies" -> "Let $\eta$ satisfy"
Page 14, equation (38), the last term should be part of the sum rather than be written separately from it
Page 15, the step where $\sum_{\substack{\tau'=0 \\ \tau'\neq \tau}}^{t-1} \rho^{\frac{t-\tau'}{2}}$ is changed to $\sum_{\tau'=0}^{t-1} \rho^{\frac{t-\tau'}{2}} - \rho^{\frac{t-\tau}{2}}$ should be identity rather than inequality
Page 16, "where we used assumptions 1-3" -> "where we used Assumptions 1-3"
Page 17, "$F(\cdot)$ is Lipschitz smooth" -> "$F(\cdot)$ is $L$-smooth" or "$F(\cdot)$ has Lipschitz gradient"
Page 18, equation (52), the last terms in the right-hand sides are detached from the sums, use () to make them part of the summation

**Strengths And Weaknesses:**

## Strengths
First of all, I want to state that I checked all proofs and I did not find any serious mistake. I think the proofs are correct.
I also found the work to be sufficiently well-written. There are some typos, but they are fixable. The only missing piece in terms of writing is a good discussion of the paper's limitations.
Finally, I found the topic of the work to be quite meaningful. There is an ongoing effort to make decentralized optimization useful in applications and I think the combination of decentralization and distributional robustness makes sense.

I do not have much to say about the experiments, but it seems they are sufficiently good. For some reason, the authors only compared to DSGD, so there is no comparison to non-decentralized algorithms, but the paper seems to be primarily theoretical, so I will not request any additional experiments.

## Weaknesses
The theory in this work is far from satisfying. There are several major issues:
1. First of all, it is not exactly clear if the authors aim at solving problem (5) or (6). From the text, it appears that the main interest is in having robustness as given in formulation (5). However, there are no guarantees for that problem in this paper. Instead, Theorem 1 states convergence for the gradient norm of the regularized objective. The authors should either explain how these guarantees translate to guarantees for problem (5), or give sufficient motivation to solve (6).
2. Secondly, I am quite unhappy about the results. The error in Theorem 1, $O\left(\frac{1}{T} + \frac{1}{\mu^2} \right)$, does not go to 0 if we increase the number of iterations, so it does not guarantee convergence. It could be fine if we could also increase $\mu$, but the issue is that in this case the problem would get closer and closer to the standard empirical risk minimization. So, the more robustness we want the worse is the guarantee, which I find to be a very significant limitation. Resolving this issue is absolutely necessary for the paper to be accepted.
3. Next, I am a bit unhappy about Assumption 2. The assumption requires almost surely bounded gradient gradients and bounded loss values, which seems to be quite restrictive. This type of assumption has been criticised in the literature on federated learning (see Khaled et al., "Tighter theory for local SGD on identical and heterogeneous data").
4. I am also concerned about implicit dependence of "constants" on $\mu$. For instance, equation (12) in Assumption 1, equation (14) in Assumption 2, and equation (17) in Assumption 3 all state the bounds in terms of universal constants even though they involve $\mu$. It seems that the values of $L_2$, $G_2$ and $\sigma_3$ implicitly depend on $\mu$, and at least in Corollary 1, the value of $\mu$ is not assumed to be fixed. The authors should either justify why these constants do not depend on $\mu$, or they should fix the assumptions and make the necessary corrections in the theoretical results.
5. Corollary 1 is very strange, if not incorrect. First of all, why can we take $\eta=\sqrt{\frac{K}{T}}$? It is not apparent at all why this value would satisfy the conditions of Theorem 1. Secondly, the value $\mu = \left(\frac{1}{KT}\right)^\frac{1}{4}$ does not lead to the upper bound as written in (27). Indeed, $\mu^2$ is in the denominator in (26), so to obtain the expression in (27), one needs to set $\mu=\left(KT\right)^\frac{1}{4}$, which to me makes no sense.

---

> ### Author Response · Authors · 2022-06-03
> **Reply to Reviewer Xb1C (Part 1)**
>
> ## Weaknesses:
> **Comment:**
> >First of all, it is not exactly clear if the authors aim at solving problem (5) or (6). From the text, it appears that the main interest is in having robustness as given in formulation (5). However, there are no guarantees for that problem in this paper. Instead, Theorem 1 states convergence for the gradient norm of the regularized objective. The authors should either explain how these guarantees translate to guarantees for problem (5), or give sufficient motivation to solve (6).
>
> **Reply:** We thank the reviewer for the comment. First, we would like to mention that we focus on solving (6) rather than (5), since solving (5) in a decentralized manner is a challenging task as stated in the paper. In fact, the focus is on solving (8), which follows from (6) after using the KL divergence. Introducing (5) makes the storyline and derivations more coherent, which explains the transition from the ERM to DRO formulation.
>
> ---
> **Comment:**
> >Secondly, I am quite unhappy about the results. The error in Theorem 1, $\mathcal{O}\left(\frac{1}{T} + \frac{1}{\mu^2} \right)$, does not go to 0 if we increase the number of iterations, so it does not guarantee convergence. It could be fine if we could also increase $\mu$, but the issue is that in this case the problem would get closer and closer to the standard empirical risk minimization. So, the more robustness we want the worse is the guarantee, which I find to be a very significant limitation. Resolving this issue is absolutely necessary for the paper to be accepted.
>
> **Reply:** We thank the reviewer for the comment. On one hand, we would like to mention that in the convergence analysis of the DSGD in (Lian et al., 2017, Theorem 1), the rate does not go to 0 if we increase the number of iterations. This is similar to the work of  (Wang et al., 2019, Theorem 2) where the error does not go to 0, if we increase the number of iterations. In both cases, the rate of $\mathcal{O}\left(\frac{1}{T} + \frac{1}{\sqrt{KT}} \right)$ is recovered once the learning rate is chosen to be $\eta \sim \sqrt{\frac{K}{T}}$. On the other hand, the robustness-accuracy tradeoff, i.e. the more robustness we want the worse is the guarantee in terms of accuracy, is a well-known tradeoff and has been noted in several works (refer to [[Ref1](https://proceedings.mlr.press/v119/raghunathan20a.html), Section 5], [[Ref2](https://arxiv.org/abs/2002.10716), Table 2], and [[Ref3](https://proceedings.mlr.press/v97/zhang19p.html), Table 4]) as well as in Table 1 in our work.
>
> ---
> **Comment:**
> >Next, I am a bit unhappy about Assumption 2. The assumption requires almost surely bounded gradient gradients and bounded loss values, which seems to be quite restrictive. This type of assumption has been criticised in the literature on federated learning (see Khaled et al., "Tighter theory for local SGD on identical and heterogeneous data").
>
> **Reply:** We agree with the reviewer that the bounded gradient assumption is rather strong. However, we would like to mention that such an assumption has been widely used in previous works focusing on robustness in FL, e.g. Mohri et al., 2019; Li et al., 2020d; and Deng et al., 2020, and is needed for the current convergence analysis. The relaxation of the uniform boundedness of the gradient is left for future work.
>
> ---
> **Comment:**
> >I am also concerned about implicit dependence of "constants" on $\mu$. For instance, equation (12) in Assumption 1, equation (14) in Assumption 2, and equation (17) in Assumption 3 all state the bounds in terms of universal constants even though they involve $\mu$. It seems that the values of $L_2$, $G_2$, and $\sigma_3$ implicitly depend on $\mu$, and at least in Corollary 1, the value of $\mu$ is not assumed to be fixed. The authors should either justify why these constants do not depend on $\mu$, or they should fix the assumptions and make the necessary corrections in the theoretical results.
>
> **Reply:** We thank the reviewer for the comment. The local loss function in (8) are defined as $F_i(\theta) = h \circ f_i(\theta)$, where $h(\theta) = \exp\big(\theta/\mu\big)$. Note that for the convergence analysis, we require to make assumptions on both functions $h$ and $f_i$. We used similar assumptions to (Wang et al., 2016; 2017; Li et al., 2020e; Huang et al., 2021). The only difference is that we wrote these assumptions explictily using the explicit expression of $h$. Hence, these assumptions hold for any value of $\mu$ and the ''constants'' do not depend on $\mu$. Furthermore, the value of the parameter $\mu$ in Corollary 1 is assumed to be fixed, i.e. $\mu=\left(KT\right)^\frac{1}{4}$.

---

> > ### Comment · Reviewer_Xb1C · 2022-06-03
> > **Discussion**
> >
> > Thank you for the response.
> >
> > **On distributional robustness**
> > > we focus on solving (6) rather than (5), since solving (5) in a decentralized manner is a challenging task as stated in the paper. In fact, the focus is on solving (8), which follows from (6) after using the KL divergence.
> >
> > Thank you for the clarification. Unlike problem (5), problems (6) and (8) are relaxed formulations, so can you please explain if solving problems (6) and (8) has any guarantees for distributional robustness? You mention in the contributions that you "propose a distributionally robust learning algorithm". How is this claim supported?
> >
> > **The error in other works does go to 0**
> > > in the convergence analysis of the DSGD in (Lian et al., 2017, Theorem 1), the rate does not go to 0 if we increase the number of iterations.
> >
> > **This is not true**, see Corollary 2 in (Lian et al., 2017). The only thing they need to do is to decrease the stepsize, which makes sense (and I do not have issues about the terms related to the stepsize in your bounds either).
> >
> > > This is similar to the work of (Wang et al., 2019, Theorem 2) where the error does not go to 0.
> >
> > This is not similar because (Wang et al., 2019) propose a new algorithm FedNova, **whose error does go to 0, as stated in their Theorem 3**. Their Theorem 2 is about a family of algorithms satisfying identity (4) in their work, which they argue to be suboptimal. Thus, comparing to their Theorem 2 is not meaningful.
> >
> > Since your bound has an extra term $O(\frac{1}{\mu^2})$ in it, which does not go to 0, it means that, as far as I understand, your algorithm does not solve neither of the problems (5), (6), or (8). The error would go to 0 only if we increase $\mu$, which corresponds to solving the objective without distributional robustness. Hence, I have to ask: why Theorem 1 is meaningful?
> >
> > **The dependence on $\mu$**
> > >  The only difference is that we wrote these assumptions explictily using the explicit expression of $h$. Hence, these assumptions hold for any value of  and the ''constants'' do not depend on $\mu$.
> >
> > I cannot agree with any assumptions on boundedness of the exponent function, it makes no sense even if it was done in prior works. Your inequality (14) clearly states that $\exp(\ell(\theta, \xi_i)/\mu)\le G_2$, which is equivalent to saying that $G_2 = \sup_{\theta, \xi_i}\exp(\ell(\theta, \xi_i)/\mu)$. Clearly, the right hand side depends on $\mu$, so $G_2$ must depend on $\mu$ as well. The results are not meaningful if this is not taken into account.
> >
> > > Furthermore, the value of the parameter  in Corollary 1 is assumed to be fixed, i.e, $\mu=(KT)^{\frac{1}{4}}$
> >
> > If $\mu$ depends on $T$, it is not fixed. "Fixed" means that its value does not change when we run the algorithm for more iteration.

---

> > > ### Author Response · Authors · 2022-06-06
> > > **Reply to discussion**
> > >
> > > **Comment:**
> > > >Unlike problem (5), problems (6) and (8) are relaxed formulations, so can you please explain if solving problems (6) and (8) has any guarantees for distributional robustness? You mention in the contributions that you "propose a distributionally robust learning algorithm". How is this claim supported?
> > >
> > > **Reply:** We thank the reviewer for the comment. Indeed, problem (6) is a regularized formulation of (5); yet it still solves the distributional robust optimization problem while imposing a regularizer on $\lambda$ captured by the regularization function $\phi(\lambda,1/K)$. This has been considered in (Mohri et al, 2019, Section 5.1) and (Deng et al, 2020, Section 5). As stated in our response, introducing (5) is to make the storyline coherent. If the reviewer thinks that this makes the reader confused, we can simply state that we are interested in solving the regularized distributional robust optimization problem given in (6) without mentioning (5).
> > >
> > > ---
> > > **Comment:**
> > > >This is not true, see Corollary 2 in (Lian et al., 2017). The only thing they need to do is to decrease the stepsize, which makes sense (and I do not have issues about the terms related to the stepsize in your bounds either).
> > >
> > > **Reply:** Corollary 2 in (Lian et al., 2017) chooses the hyperparameter $\eta$ such that the error goes to 0. Note that the error in Theorem 1 in (Lian et al., 2017) does not go to 0 unless $\eta$ is chosen as in Corollary 2. Similarly, we select the values of the hyperparameters $\eta$ and $\mu$, in our Corollary 1, to ensure that the error goes to 0.
> > >
> > > ---
> > > ---
> > > **Comment:**
> > > >This is not similar because (Wang et al., 2019) propose a new algorithm FedNova, whose error does go to 0, as stated in their Theorem 3. Their Theorem 2 is about a family of algorithms satisfying identity (4) in their work, which they argue to be suboptimal. Thus, comparing to their Theorem 2 is not meaningful. Since your bound has an extra term $O(\frac{1}{\mu^2})$ in it, which does not go to 0, it means that, as far as I understand, your algorithm does not solve neither of the problems (5), (6), or (8). The error would go to 0 only if we increase $\mu$, which corresponds to solving the objective without distributional robustness. Hence, I have to ask: why Theorem 1 is meaningful?
> > >
> > > **Reply:** The paper we are referring to in our response is [(Wang et al., 2019)](https://arxiv.org/pdf/1905.09435.pdf). The algorithm is called MATCHA. We are not comparing to the paper mentioned above. We are simply stating that similar convergence analyses in the DSGD literature exist, in the sense that unless the learning rate is chosen to be of a specific form, the error does not go to 0 in these analyses.
> > >
> > > ---
> > > **Comment:**
> > > >I cannot agree with any assumptions on boundedness of the exponent function, it makes no sense even if it was done in prior works. Your inequality (14) clearly states that $\exp(\ell(\theta, \xi_i)/\mu)\le G_2$, which is equivalent to saying that $G_2 = \sup_{\theta, \xi_i}\exp(\ell(\theta, \xi_i)/\mu)$. Clearly, the right hand side depends on $\mu$, so $G_2$ must depend on $\mu$ as well. The results are not meaningful if this is not taken into account.
> > >
> > > **Reply:** Thank you for the follow-up explanation. We would like to mention that inequality (14) is written mathematically as:
> > > \begin{align}
> > > \exists G_2 > 0 \text{ such that } \forall \theta \in \mathbb{R}^d, \forall \mu \in \mathbb{R}_{+}, ~\exp(\ell(\theta, \xi_i)/\mu)\le G_2.
> > > \end{align}
> > > In other words, $G_2$ does not depend on $\mu$ (existence of $G_2$ comes before the statement holding for every value of $\mu$). Note that, for example, this can be fulfilled using the value of $\mu$ from Corollary 1 (in this case $\mu \geq 1$) and assuming that the range of the loss function is bounded, i.e. $\ell(\theta, \xi_i) \leq M$ (refer to our response to reviewer 7oCx, part 2, first comment). In fact, in this case, we have that:
> > > \begin{align}
> > >     \exp(\ell(\theta, \xi_i)/\mu) \leq \exp(\ell(\theta, \xi_i)) \leq \exp(M) := G_2
> > > \end{align}
> > >
> > > ---
> > > **Comment:**
> > > >If $\mu$ depends on $T$, it is not fixed. "Fixed" means that its value does not change when we run the algorithm for more iteration.
> > >
> > > **Reply:** Our understanding of what the term "fixed" means was that the parameter changes every iteration, which is not the case as previously explained. We thank the reviewer for explaining he/she means by "fixed". We would like to mention that the regularization parameter $\mu$ is a hyperparameter similar to the learning rate $\eta$. In that sense, we do not see an issue if it depends on the number of iterations $T$.

---

> > > > ### Comment · Reviewer_Xb1C · 2022-06-06
> > > > **More clarification needed**
> > > >
> > > > Thank you for responses. I feel like I am getting closer to a conclusion about this work, which I will then try to discuss with other reviewers. A few more clarifications would still help me:
> > > > > **Reply**: We thank the reviewer for the comment. Indeed, problem (6) is a regularized formulation of (5); yet it still solves the distributional robust optimization problem while imposing a regularizer on $\lambda$ captured by the regularization function $\phi(\lambda, 1/K)$. This has been considered in (Mohri et al, 2019, Section 5.1) and (Deng et al, 2020, Section 5). As stated in our response, introducing (5) is to make the storyline coherent. If the reviewer thinks that this makes the reader confused, we can simply state that we are interested in solving the regularized distributional robust optimization problem given in (6) without mentioning (5).
> > > >
> > > > I understand that the problems are connect, but can you provide any measure of how connected they are? For instance, is the solution of (6) close to the solution of (5)? It would be nice to characterize the difference between the problems in terms of $\mu$, are the authors aware of any result like that?
> > > >
> > > > > **Reply**: Corollary 2 in (Lian et al., 2017) chooses the hyperparameter $\eta$ such that the error goes to 0. Note that the error in Theorem 1 in (Lian et al., 2017) does not go to 0 unless $\eta$ is chosen as in Corollary 2. Similarly, we select the values of the hyperparameters $\eta$ and $\mu$, in our Corollary 1, to ensure that the error goes to 0.
> > > >
> > > > Exactly. What I am trying to say is that it is ok to choose $\eta$ any way you want because it is part of the algorithm. However, $\mu$ is part of the objective, so it is not ok to choose it for your convenience.
> > > >
> > > > > **Reply**: The paper we are referring to in our response is (Wang et al., 2019). The algorithm is called MATCHA. We are not comparing to the paper mentioned above. We are simply stating that similar convergence analyses in the DSGD literature exist, in the sense that unless the learning rate is chosen to be of a specific form, the error does not go to 0 in these analyses.
> > > >
> > > > I understand that the authors have to choose $\eta$ to make the error go to 0, this is fine. I am only concerned about the choice of $\mu$.
> > > >
> > > > > **Reply**: Thank you for the follow-up explanation. We would like to mention that inequality (14) is written mathematically as:
> > > > $\exists G_2 > 0 \textrm{ such that } \forall \theta\in\mathbb{R}^d, \forall \mu \in \mathbb{R}_+, \exp(\ell(\theta, \xi_i)/\mu)\le G_2$
> > > >
> > > > This is not possible unless we assume a lower bound on $\mu$ because as $\mu\to 0$ it holds $\exp(\ell(\theta, \xi_i)/\mu)\to +\infty$. Since small values of $\mu$ are those that make problem (6) closer to (5), it seems to me that making any assumption in the form $\mu\ge 1$ is not alright.

---

> > > > > ### Author Response · Authors · 2022-06-07
> > > > > **Reply (Part 1)**
> > > > >
> > > > > **Comment:**
> > > > > > I understand that the problems are connect, but can you provide any measure of how connected they are? For instance, is the solution of (6) close to the solution of (5)? It would be nice to characterize the difference between the problems in terms of , are the authors aware of any result like that?
> > > > >
> > > > > **Reply:** We thank the reviewer for the followup questions. In what follow, we try to answer the question, to the best of our knowledge. According to [Qian et al, 2019](https://arxiv.org/pdf/1805.07588.pdf), problem (6) can be re-written using the duality theory as
> > > > > \begin{align}
> > > > >     \min_{\Theta \in \mathbb{R}^d} \max_{\lambda \in \Delta} \sum_{i=1}^K \lambda_i f_i(\Theta) \\
> > > > >     s.t. ~ \phi(\lambda,1/K) \leq \tau
> > > > > \end{align}
> > > > > where for every $\mu$, we associate a $\tau$ value. From [Namkoong et al, 2016](https://papers.nips.cc/paper/2016/file/4588e674d3f0faf985047d4c3f13ed0d-Paper.pdf) and [Duchi et al., 2016](https://arxiv.org/abs/1610.03425), for \textbf{convex} loss functions, we have that
> > > > >
> > > > > \begin{align}
> > > > >  \max_{\lambda \in P_{\rho,K}} \sum_{i=1}^K \lambda_i \ell_i(\Theta) = \frac{1}{K} \sum_{i=1}^K  \ell_i(\Theta) + \sqrt{\tau Var_{P_0}[\ell(\Theta, \xi)]} + o_{P_0}(K^{-\frac{1}{2}}),
> > > > > \end{align}
> > > > >
> > > > > where $\ell(\Theta) = [\ell_1(\Theta), \dots, \ell_K(\Theta)]^T \in \mathbb{R}^K$ is the vector of losses, $P_0$ is the empirical probability distribution,  and the ambiguity set $P_{\rho,K}$ is defined as $P_{\rho,K} = \[ \lambda \in \mathbb{R}^K, \sum_{i=1}^K \lambda_i = 1, \lambda \geq 0, \phi(\lambda,1/K) \leq \tau \]$.
> > > > >
> > > > > Note that problem (5) is equivalent to the case when $\tau = 0$. Thus, we can link the objective of problem (6) to (5) as
> > > > > \begin{align}
> > > > >     \min_{\Theta \in \mathbb{R}^d} \max_{\lambda \in P_{\rho,K}} \sum_{i=1}^K \lambda_i f_i(\Theta) = \min_{\Theta \in \mathbb{R}^d} \max_{\lambda \in P_{0,K}} \sum_{i=1}^K \lambda_i f_i(\Theta) +  \sqrt{\tau Var_{P_0}[\ell(\Theta, \xi)]}
> > > > > \end{align}

---

> > > > > > ### Comment · Reviewer_Xb1C · 2022-06-07
> > > > > > **Thanks**
> > > > > >
> > > > > > Thank you for providing these results. It is still not exactly clear to me how $\tau$ is connected to $\mu$, but at least the impact of regularization can be measured.

---

> > > > > ### Author Response · Authors · 2022-06-10
> > > > > **Reply (Part 2)**
> > > > >
> > > > > **Comment:**
> > > > > > Exactly. What I am trying to say is that it is ok to choose $\eta$ any way you want because it is part of the algorithm. However, $\mu$ is part of the objective, so it is not ok to choose it for your convenience.
> > > > >
> > > > > **Reply:** We thank the reviewer for the explanation. We updated the proof in the current manuscript where now, in Corollary 1, we don't specify the value of $\mu$, but we choose the value of the hyperparameters $\eta$ and $B$.

---

> > > > > > ### Comment · Reviewer_Xb1C · 2022-06-10
> > > > > > **This is a great improvement!**
> > > > > >
> > > > > > I thank the authors for making this change, my concern about the choice of $\mu$ is fully resolved.

---

> ### Author Response · Authors · 2022-06-03
> **Reply to Reviewer Xb1C (Part 2)**
>
> **Comment:**
> >Corollary 1 is very strange, if not incorrect. First of all, why can we take $\eta=\sqrt{\frac{K}{T}}$? It is not apparent at all why this value would satisfy the conditions of Theorem 1. Secondly, the value $\mu = \left(\frac{1}{KT}\right)^\frac{1}{4}$ does not lead to the upper bound as written in (27). Indeed, $\mu^2$ is in the denominator in (26), so to obtain the expression in (27), one needs to set $\mu=\left(KT\right)^\frac{1}{4}$, which to me makes no sense.
>
> **Reply:** We thank the reviewer for pointing out the typo. Indeed, the value of $\mu$ is rather $\mu=\left(KT\right)^\frac{1}{4}$, and this has been fixed in the revised manuscript. The aim of Corollary 1 is to see, under which choice of the hyperparameters $\eta$ and $\mu$, the rate $\mathcal{O}(1/\sqrt{KT})$ is recovered. A similar reasoning has been used in (Lian et al., 2017 and (Wang et al., 2019). We admit that the choice of $\mu$ presents a limitation of the work. Moreover, we would like to mention that in practice, even with the smaller values of $\mu$, the algorithm still converges, kindly refer to Section 6.4 where we study the impact of $\mu$ on average and worst test accuracies. Finally, we included a discussion section on the limitations of this work in the revised manuscript, where the choice of $\mu$ is one of them.
>
> ---
> ## Minor points:
> **Comment:**
> >One thing that really confuses me is that superscripts are used both as iteration counters and matrix powers, sometimes within the same equation, for instance in (36).
>
> **Reply:** We thank the reviewer for his comment. We think that this could be understood from the context. If the variable in question does not depend on the iteration counter, then the superscipt is used to denote the power, e.g. the case of the matrix $W$.
>
> ---
> **Comment:**
> >The authors claim several times that their algorithm is "robust to data heterogeneity". I am not sure what this means and where it is shown. Instead, it seems that the algorithm is robust to distribution shift, which is not the same as robust to higher data heterogeneity. Moreover, I suspect that any algorithm that tries to use distributional robustness would be less robust to data heterogeneity, as it may largely overfit to the data on a single client.
>
> **Reply:** We agree with the reviewer's comment. In the revised manuscript, we fix this claim by stating that the algorithm is rather robust to distribution shift (one form of data heterogeneity) and not to higher data heterogeneity.
>
> ---
> **Comment:**
> >Why is DR-DSGD better than DSGD in the experiments in terms of average test accuracy? From what I understand, DR-DSGD is solving a harder problem, so it is natural to expect that it would be better in terms of worst-distribution test accuracy, but the improvement of the average accuracy is quite surprising. For instance, in the work on "tilte" risk minimization, Table 7, only an improvement for the worst devices is reported.
>
> **Reply:** We thank the reviewer for pointing out this point. Upon investigating this behaviour, we found a ``bug'' in the computation of the average test accuracy for DSGD. We fixed this issue in the revised manuscript. We can now clearly see that DR-DSGD has almost a similar performance as DSGD in terms of the average test accuracy.
>
> ---
> ## Typos:
> **Reply:** We thank the reviewer for taking the time to point out these typos. We fixed all of them in the revised manuscript.

---

> > ### Comment · Reviewer_Xb1C · 2022-06-03
> > **One question not answered**
> >
> > Could you please also explain why choosing $\eta = \sqrt{\frac{K}{T}}$ is allowed? As far as I can see from Theorem 1, the value of $\eta$ should not exceed $\frac{1}{L}$, which does not seem to be taken into account.

---

> > > ### Author Response · Authors · 2022-06-06
> > > **Reply**
> > >
> > > We thank the reviewer for pointing this out. To satisfy the condition in Theorem 1, we change the choice of $\eta$ to $\eta = \frac{1}{2L + \sqrt{T/K}}$.

---

> ### Comment · Reviewer_Xb1C · 2022-06-11
> **A question about equation (57)**
>
> I realized that there is one thing that I do not completely understand. Namely, the transition from equation (56) to (57) seems to consist of applying four different upper bounds and one of them is unclear to me. I think I understand what happens to the last two sums in the right-hand side of (56): the first one of them is upper bounded using bounded variance and the fact that $\mathbb{E}[g_i(\overline {\theta^t})]=\nabla f_i(\overline{\theta^t})$; the second of them is bounded using Lipschitzness, $\Vert g_i(\overline{\theta^t}) - g_i(\theta_i^t)\Vert \le L\Vert\overline{\theta^t}  - \theta_i^t\Vert$.
>
> Can the authors please explain what happens to the following two quantities between (56) and (57):
> $$
>     \left|\exp\left(\frac{f_i(\overline{\theta^t})}{\mu}  \right) - h(\overline{\theta^\tau}; \mu)) \right|
> $$
> and
> $$
>     \left|h(\overline{\theta^\tau});\mu)  - h({\theta_i^\tau}; \mu)) \right|?
> $$
> As far as I can see, $h(\overline{\theta^\tau}; \mu))$ is not an unbiased estimate of $\exp\left(\frac{f_i(\overline{\theta^t})}{\mu}  \right)$. Therefore, I am not sure why $B$ shows up in the denominator in (57). I would really appreciate a detailed explanation of this step.
>
> By the way, what is $\tau$ in (54)-(56), shouldn't it be $t$?

---

> > ### Author Response · Authors · 2022-06-13
> > **Reply**
> >
> > **Comment:**
> > > Transition from (56) to (57)
> >
> > **Reply:** We thank the reviewer for the question. In the revised manuscript, we detail the steps for the transition from (56) to (57).
> >
> > ---
> > **Comment:**
> > > By the way, what is $\tau$ in (54)-(56), shouldn't it be $t$?
> >
> > **Reply:** We thank the reviewer for pointing this out. We corrected this typo in the current version.

---

> > > ### Comment · Reviewer_Xb1C · 2022-06-13
> > > **Thanks, can you clarify one statement?**
> > >
> > > I thank the authors for providing a detailed explanation of the transition. Can you please explain why in the added explanation you wrote that for $j\neq i$ it holds $\mathbb{E}[\langle X_i, X_j\rangle] = 0$? As far as I can see, $\mathbb{E}[X_i]\neq 0$ because in general $\mathbb{E}[\exp(Y)]\neq \exp(\mathbb{E}[Y])$, where $Y$ is a random variable.

---

> > > > ### Author Response · Authors · 2022-06-17
> > > > **Reply**
> > > >
> > > > We thank the reviewer for the question. We agree with the remark. There was a typo in the definition of the function $h(\theta;\mu)$. We modified the proof based on this correction. We hope that the steps are clearer now.

---

> > > > > ### Comment · Reviewer_Xb1C · 2022-06-17
> > > > > **I'm still confused**
> > > > >
> > > > > Dear authors, I still do not see why you claim that $\mathbb{E}[\langle X_i^t, X_j^t\rangle]=0$ between equations (57) and (58). I understand that $\mathbb{E}\left[\frac{\ell(\overline {\theta^t}, \xi_i^t)}{\mu}\right] = \frac{f_i(\overline {\theta^t})}{\mu}$, but the exponent cannot be exchanged with expectation, so
> > > > > $$
> > > > > \mathbb{E}[X_i^t]=\mathbb{E}\left[\exp\left(\frac{\ell(\overline {\theta^t}, \xi_i^t)}{\mu} \right)\right] - \exp\left(\frac{f_i(\overline {\theta^t})}{\mu} \right) \neq 0.
> > > > > $$
> > > > >
> > > > > Right now, the proof looks wrong to me.

---

> > > > > > ### Author Response · Authors · 2022-06-17
> > > > > > **Reply**
> > > > > >
> > > > > > We changed the definition of $X_i$ in the revised manuscript. We no longer exchange the exponent with expectation.

---

> > > > > > > ### Comment · Reviewer_Xb1C · 2022-06-17
> > > > > > > **Thanks**
> > > > > > >
> > > > > > > Now it looks correct.